# Counterfactual Explanations for Time Series Data via Reinforcement Learning

## Abstract

Counterfactual (CF) explanations are a powerful tool in Explainable AI (XAI), providing actionable insights into how model predictions could change under minimal input alterations. Generating CFs for time series, however, remains challenging: existing optimization-based methods are often instance-specific, impose restrictive constraints, and struggle to ensure both validity and plausibility. To address these limitations, we propose a reinforcement learning (RL) framework for counterfactual explanation in time series. Our actor–critic agent learns a policy in the latent space of a pre-trained autoencoder, enabling the generation of counterfactuals that balance validity and plausibility without relying on rigid handcrafted constraints. Once trained, the RL agent produces counterfactuals in a single forward pass, ensuring scalability to large datasets. Experiments on diverse benchmarks demonstrate that our approach generates valid and plausible counterfactuals, offering a reliable alternative to existing methods.

## 1 Introduction

Understanding the predictions of machine learning models is especially important in high-stakes areas such as healthcare monitoring, financial risk assessment, and space weather forecasting (Loh et al., 2022; Bharati et al., 2023; Černevičienė & Kabašinskas, 2024; Bussmann et al., 2020; Camps-Valls et al., 2025). In these domains, decisions informed by models can have significant consequences for human health, safety, or large-scale operations. While modern machine learning models, particularly deep neural networks, achieve remarkable accuracy, their black-box nature raises concerns about transparency, accountability, and trustworthiness (Dwivedi et al., 2023). Stakeholders in sensitive applications need more than predictions — they also need to understand why a decision was made and what factors influenced it.

Counterfactual explanations (CFEs) address this by asking a precise "what-if" question: what minimal, plausible changes to the input would flip the model's prediction (Wachter et al., 2017)? By specifying the smallest set of feature changes sufficient to alter the outcome, CFEs indirectly reveal which features are most influential and can be mapped to actionable recourse—that is, concrete steps a user could take to pursue a different decision. For example, in a loan application scenario, a CFE might show that if the applicant's annual income were slightly higher or their credit utilization ratio slightly lower, the model would have approved the loan instead of rejecting it. Such insights both clarify model reasoning and provide applicants with meaningful guidance for improving future outcomes.

While counterfactuals have been widely studied in tabular and image domains (Wachter et al., 2017; Mothilal et al., 2020; Looveren & Klaise, 2021; Guidotti, 2024; Verma et al., 2024), developing meaningful CFEs for time series remains comparatively less explored. For images, counterfactual changes are often intuitive and directly interpretable: a small alteration to a shape, color, or texture can be visually inspected, and the plausibility of the change can be judged at a glance. For tabular data, constraints on features are also relatively straightforward to define—certain attributes like age or gender are immutable, while others, such as income or credit score, can be adjusted within realistic ranges. This makes it possible to specify per-feature rules and ensure that counterfactuals stay valid and plausible as long as they respect these ranges and dependencies. Time series data, however, fundamentally differ from tabular data in ways that make counterfactual generation far more challenging. Unlike tabular features, which describe static attributes that can be modified indepen-

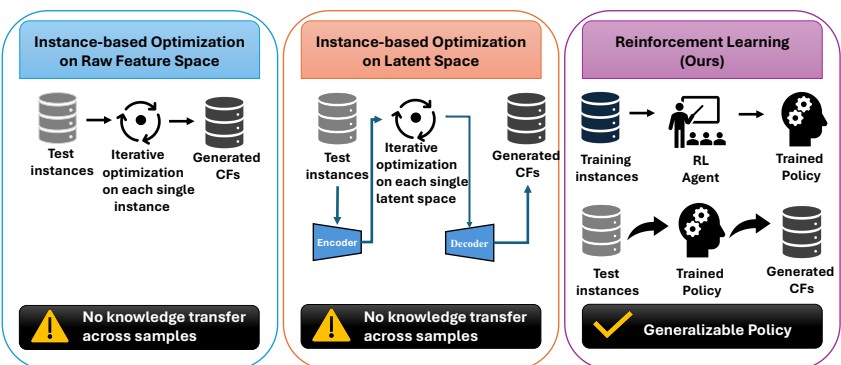

Figure 1: Comparison of three counterfactual generation paradigms.

dently under simple constraints, each feature in a time series is a temporal trajectory that evolves over time. Any modification must therefore respect underlying temporal dependencies—such as trends, local continuity, and recurring patterns—rather than treating each time step as an isolated value. A small perturbation at a single point may disrupt these dynamics, creating unnatural jumps or breaks that no longer resemble a realistic sequence. As a result, plausibility cannot be ensured through straightforward feature-level rules, as in tabular data; instead, it depends on whether the entire modified sequence still forms a coherent and temporally consistent signal. This makes defining and enforcing plausibility in time series counterfactuals significantly more difficult. Naive perturbations—such as locally modifying a short segment (Delaney et al., 2021; Li et al., 2022b) often disrupt global temporal structure and produce unrealistic patterns. More advanced optimization-based approaches (Filali Boubrahimi & Hamdi, 2022; Li et al., 2022a; Hosseinzadeh et al., 2024; Wang et al., 2024) attempt to mitigate this by minimizing combinations of distance losses, prediction losses, and smoothness constraints to maintain temporal continuity and avoid unrealistic fluctuations. However, those mainstream approaches are still fundamentally *instance-based*: whether operating directly in the raw feature space or in a learned latent space, they run a separate iterative optimization loop for each input and do not transfer knowledge across samples. As a result, they are slow and computationally expensive for large or high-dimensional datasets, and their updates may still lead to implausible results because the optimization is driven by what works for a single instance rather than by patterns learned at the dataset level. Together, these challenges make it difficult to balance plausibility and efficiency, limiting the practicality of current counterfactual methods for time series at scale.

To address these limitations, we propose a reinforcement learning (RL) framework that learns from experience rather than optimizing each instance in isolation. As illustrated in Figure 1, mainstream approaches—whether operating in the raw feature space or in a latent space—conduct iterative optimization separately for every input and do not transfer knowledge across samples. In contrast, our RL framework interacts with many training examples and gradually acquires a transferable policy that captures general patterns useful for generating counterfactuals. Through this experience-driven learning process, the agent learns perturbation strategies that generalize across instances, allowing it to produce counterfactuals for new inputs with a single forward pass. This improves scalability and provides a more stable and reusable mechanism for generating valid and realistic counterfactuals at scale.

The contributions of this paper are as follows: 1) we introduce an RL-based framework that learns from experience rather than performing per-instance optimization. By acquiring a transferable policy that can be applied to new inputs with a single forward pass, the method supports efficient batch generation of counterfactuals and offers greater scalability than mainstream instance-based approaches; 2) instead of applying explicit constraints to raw time series data, we perturb in the latent space and use an L1+L2 distance penalty to keep the counterfactuals close to the original input, which helps maintain realistic structure; 3) the approach is model-agnostic, requiring only classifier predictions, allowing it to operate with black-box classifiers without gradient access; 4) our approach opens the door to leveraging RL techniques (e.g., policy learning, exploration strategies) for explanation, offering a more flexible alternative to handcrafted optimization objectives.

## 2  RELATED WORK

Counterfactual explanations (CFEs) have received increasing attention because of their ability to provide intuitive and actionable insights into machine learning predictions. By identifying minimal changes that alter a model's decision, CFEs help users understand model behavior and explore alternative outcomes. Such explanations are especially valuable in high-stakes applications, where transparency and interpretability are as important as accuracy.

Early efforts in time series CFEs relied on instance-based heuristics, producing counterfactuals by perturbing each instance individually, often through segment replacement. For example, NG (Delaney et al., 2021) and MG-CF (Li et al., 2022b) construct counterfactuals by altering or replacing salient subsequences drawn from the data (e.g., CAM-highlighted regions, class motifs). Their main strength is interpretability, since the modifications are grounded in real observed patterns that domain experts can readily relate to. In addition, because they reuse real subsequences, these methods often produce results that appear plausible at the local level. However, they depend heavily on the quality of saliency or motif mining and can struggle with validity (failing to consistently flip the label) and global coherence (as a swapped segment may not integrate smoothly with the rest of the sequence), which reduces their robustness across datasets.

Building on this idea, optimization-based approaches formulate counterfactual generation as an optimization problem for each input instance. The classic formulation by Wachter et al. (2017), originally introduced for tabular data, has since been widely adopted as a baseline in time series studies (Li et al., 2022b;a; Bahri et al., 2022a). Wachter typically minimizes a loss that balances prediction validity (ensuring the label flips) with proximity to the original instance, often measured by L1 or L2 distance. While this formulation is simple and intuitive, proximity constraints alone cannot guarantee temporal plausibility, and naïve pointwise perturbations often disrupt sequential dynamics, leading to unrealistic counterfactuals. To address this limitation, constrained optimization variants have been proposed. For example, SG-CF (Li et al., 2022a) and TeRCE (Bahri et al., 2022b) restrict perturbations to subsequences defined by shapelets or temporal rules, while TimeX (Filali Boubrahimi & Hamdi, 2022) enforces temporal coherence through barycenter averaging. These constraints improve local interpretability and coherence but introduce additional rigidity, as they rely on predefined structures that may not capture global dependencies, and they still suffer from the inefficiency of per-instance optimization.

A related direction uses saliency-guided masks to localize perturbations and improve interpretability. Methods such as CELS (Li et al., 2023) and Info-CELS (Li et al., 2025) learn perturbation masks guided by gradient information and the nearest unlike neighbor, enabling sparse and interpretable modifications without the need for predefined structures or rules. These approaches produce intuitive visual explanations by highlighting learned saliency maps. However, a key limitation is their reliance on the nearest unlike neighbor: counterfactuals are generated by interpolating between the original instance and this neighbor, with the interpolation rate controlled by the saliency map. If the nearest unlike neighbor exhibits temporal shifts or misalignments, the resulting counterfactuals can still be implausible, even if the label flips successfully.

Latent-space perturbation has recently emerged as a promising direction for improving plausibility. Glacier (Wang et al., 2024) perturbs representations in both the input space and autoencoder-derived latent space, guided by saliency explainers such as LimeSegment (Sivill & Flach, 2022). The saliency vectors serve as constraints that determine which time steps should be perturbed, helping to localize modifications and preserve temporal structure. While this approach improves temporal localization, its performance remains highly sensitive to the quality of the saliency explainer. If the saliency vectors highlight noisy or irrelevant regions, the generated counterfactuals may still lack plausibility and robustness.

All of the methods discussed above are fundamentally instance-level approaches, where counterfactuals are generated independently for each input through optimization or perturbation. This limits both scalability and generalization, as the process must be repeated from scratch for every new instance. A few recent works in tabular domains have explored using reinforcement learning to overcome these limitations. For example, Samoilescu et al. (2021) formulated counterfactual generation as an RL problem in which an agent learns to apply minimal perturbations that flip a classifier's decision. Similarly, Chen et al. (2022) proposed RELAX, a model-agnostic CF generator that learns an action policy to modify tabular features, and Nguyen et al. (2021) developed a multi-agent RL

framework for generating counterfactual explanations in drug–target prediction tasks. These works demonstrate the promise of experience-driven policy learning for counterfactual generation in tabular settings. However, this perspective has been largely overlooked in the context of time series data. Motivated by this gap, we revisit the RL perspective and extend it to time series classification. Instead of optimizing each instance independently, we train an RL agent to learn a policy for generating counterfactuals by accumulating experience across many examples. Once trained, this policy can be applied directly to new inputs to generate counterfactuals in batches, providing a more scalable and generalizable alternative to instance-based optimization.

# 3 PRELIMINARY

## 3.1 COUNTERFACTUAL EXPLANATION FOR TIME SERIES CLASSIFICATION

We consider a dataset of $N$ time series $\mathcal{X} = \{x_1, \dots, x_N\}$, where each $x_i \in \mathbb{R}^{D \times T}$ represents a $D$-dimensional sequence of length $T$. Let $M : \mathbb{R}^{D \times T} \to \mathcal{Y}$ be a black-box classifier mapping each time series to one of $C$ class labels in $\mathcal{Y}$.

For an input (original) instance $x \in \mathcal{X}$, the model prediction is $y_M = M(x)$. A counterfactual instance is a perturbed version of $x$ that changes the model's decision to a user-specified target (desired) label $y_T \in \mathcal{Y}$ with $y_T \neq y_M$.

Formally, the goal is to generate a counterfactual time series $x_{CF}$ such that

$$x_{CF} = x + \delta, \quad \delta \in \mathbb{R}^{D \times T}, \quad M(x_{CF}) = y_T. \tag{1}$$

A *counterfactual explanation* in time series classification is the perturbation $\delta$ (or equivalently, the change from $x$ to $x_{CF}$) that shows how the original time series must be modified to achieve the desired prediction outcome.

## 3.2 REINFORCEMENT LEARNING (RL)

Reinforcement learning (RL) provides a general framework in which an agent learns a policy by maximizing expected rewards through interaction with an environment (Sutton et al., 1998). Although RL is commonly used for sequential decision-making, it also includes contextual bandit settings where the agent makes a single decision based on a given context and receives an immediate reward. This one-step formulation is useful in problems that require selecting an optimal action without multi-step planning.

Model-free RL methods do not require knowledge of environment dynamics. Value-based approaches learn an action-value function $Q(s, a)$, while policy-based methods directly optimize a parameterized policy. Actor–critic algorithms combine the strengths of both: the critic estimates $Q(s, a)$ and provides gradient signals, and the actor updates the policy accordingly. This structure is particularly effective in high-dimensional continuous control tasks.

Deep Deterministic Policy Gradient (DDPG) (Lillicrap et al., 2015) is a widely used actor–critic algorithm designed for continuous action spaces. The actor outputs a deterministic action $a = \mu_\theta(s)$, and the critic evaluates its value $Q_\phi(s, a)$. Training is stabilized through experience replay, which reuses past transitions, and target networks, which improve stability by smoothing the updates of both actor and critic.

# 4 METHODOLOGY

Instead of relying on instance-based optimization, where each counterfactual must be obtained by perturbing the input through the same optimization process for every instance, we employ an actor–critic reinforcement learning framework to learn a policy that generates counterfactuals efficiently. Once trained, the policy produces counterfactuals in batches through a single forward pass, offering greater scalability than per-instance optimization.

Formally, we formulate counterfactual generation as a *one-step* Markov decision process (contextual bandit):

$$\mathcal{M} = (\mathcal{S}, \mathcal{A}, R), \quad s \in \mathcal{S}, \ a \in \mathcal{A}, \ R = R(s, a), \tag{2}$$

where $\mathcal{S}$ is the state space, with $s = (z, y_M, y_T)$ representing the latent embedding $z$, current prediction $y_M$, and target class $y_T$; $\mathcal{A}$ is the action space, where $a = z_{CF}$ represents the modified latent embedding produced by the actor; and $R$ is the immediate reward function: $R(s, a) = 1$ if $M(\text{dec}(a)) = y_T$, and 0 otherwise.

Note that there is no transition function $P$ or next state $s'$: each counterfactual generation constitutes a single decision, after which the episode terminates. This differs from traditional multi-step RL, where agents navigate through sequential states. The environment in our formulation is the interaction between the time series representation and the classifier $M$. Given a state $s$, the actor generates an action $a$, which is decoded and evaluated by $M$ to produce an immediate binary reward. Unlike sequential RL problems with long-term credit assignment, our one-step formulation provides direct feedback after each decision.

To handle the high dimensionality and temporal structure of time series, we first train an autoencoder to obtain a lower-dimensional latent representation $z = \text{enc}(x)$. Operating in this compact latent space significantly reduces the effective action dimensionality for the RL agent, avoiding the need to perturb every time step of the raw sequence. Counterfactual generation is then performed in this latent space, where perturbations are smoother and more structured. The actor network $\mu_\theta$ takes the latent state $z$ as input and outputs a perturbed embedding $z_{CF}$ corresponding to a counterfactual candidate. This representation is decoded back to the input space, $\tilde{x}_{CF} = \text{dec}(z_{CF})$, which is then evaluated by the classifier to determine the reward.

We adopt a simplified deterministic actor–critic framework inspired by Deep Deterministic Policy Gradient (DDPG) (Lillicrap et al., 2015). The critic $Q_\phi(s, a)$ estimates the value of applying action $a$ in state $s$ and is trained by regressing directly to the observed (immediate) reward signal:

$$L_{\text{critic}} = \mathbb{E}\Big[\big(Q_\phi(s_t, a_t) - R_t\big)^2\Big]. \tag{3}$$

**Critic in One-Step Settings.** In standard multi-step DDPG, the critic estimates long-term cumulative rewards: $Q(s, a) = \mathbb{E}[R_t + \gamma R_{t+1} + \gamma^2 R_{t+2} + \cdots]$. However, in our one-step formulation, the critic simplifies to predicting immediate reward expectations:

$$Q(s, a) \approx \mathbb{E}[R \mid s, a] \quad \text{where } R \in \{0, 1\}. \tag{4}$$

The critic loss becomes a direct regression to observed binary rewards without bootstrapping from future values. This differs from standard DDPG's temporal-difference target: $Q_{\text{target}} = R + \gamma Q'(s', a')$. We have no $s'$ (next state) or $\gamma$ (discount factor).

Despite this simplification, the critic plays three essential roles in our framework. First, it reduces variance by learning a smooth function $Q(s, a)$ that approximates the expected reward, providing more stable gradients than the binary flip-reward signal. Second, it enables off-policy learning through experience replay, allowing stored $(s, a, R)$ tuples to be reused efficiently even as the actor policy evolves. Finally, it offers a differentiable surrogate for the non-differentiable classifier and reward, allowing the actor to be optimized via gradients of $Q(s, \mu_\theta(s))$.

The actor is trained to maximize the critic's estimate of value through the deterministic policy gradient:

$$L_{\text{max}} = -\mathbb{E}\big[Q_\phi(s_t, \mu_\theta(s_t))\big]. \tag{5}$$

To promote similarity between the counterfactual and the original input, we add a proximity regularizer that combines $L_1$ and $L_2$ distances:

$$L_{\text{prox}} = \beta \|\tilde{x}_{CF} - x\|_1 + \|\tilde{x}_{CF} - x\|_2^2. \tag{6}$$

The final actor objective is then

$$L_{\text{actor}}^{\text{total}} = L_{\text{max}} + \lambda_{\text{p}} L_{\text{prox}}. \tag{7}$$

During training, exploration is encouraged by injecting noise into the actor's proposed **perturbed embedding**, gradually transitioning from uniform random actions to noisy perturbations around the actor's output. Through this combination, the actor learns to generate counterfactuals that maximize validity while preserving proximity, and the decoder ensures that the resulting sequences remain realistic and temporally coherent. Figure 2 shows the framework of our proposed method. In the following, we explain each component of the framework, including both the training workflow and the inference workflow.

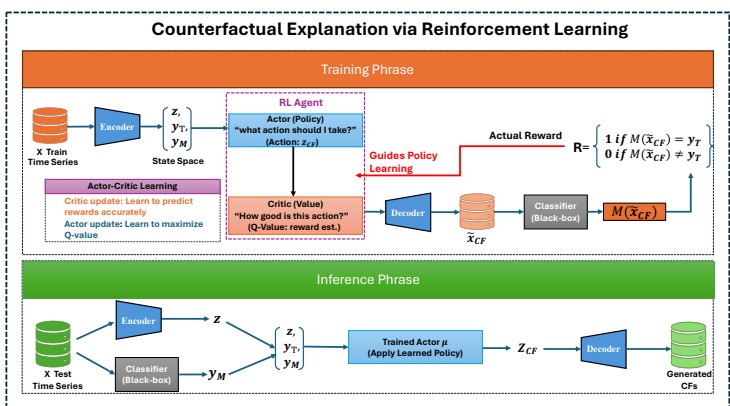

Figure 2: The overview of the proposed framework.

**Training Workflow.** Algorithm 1 summarizes the training pipeline of our reinforcement learning framework for counterfactual generation in time series, which is conducted on the training dataset. The procedure begins by loading a pre-trained encoder-decoder pair (line 1), which defines a structured latent space for applying perturbations. The actor and critic networks are initialized along with a replay buffer, and a reward function $f$ is defined to convert classifier predictions relative to the target label into a reward signal (lines 2–4).

At each training step (lines 5–12), an input time series $x$ and target label $y_T$ are sampled. The input is encoded into a latent representation $z$, and the actor generates a candidate counterfactual latent $z_{CF}$ conditioned on $(z, y_M, y_T)$. Gaussian noise is added for exploration, and the perturbed latent $\tilde{z}_{CF}$ is clipped before being decoded into a candidate $\tilde{x}_{CF}$. The black-box classifier $M$ evaluates $\tilde{x}_{CF}$ to produce a reward $R = f(M(\tilde{x}_{CF}), y_T)$, and the interaction tuple $(z, y_M, y_T, \tilde{z}_{CF}, R)$ is stored in the replay buffer.

Every $U$ steps, the framework performs a gradient update (lines 13–19). First, a minibatch from the replay buffer is used to update the critic by minimizing the squared error between its predicted value and the observed reward, thereby improving its approximation of the reward function. Next, the actor is updated with a combined objective: a policy loss, which encourages the actor to maximize the critic's Q-value and thus generate valid counterfactuals, and a proximity loss, which penalizes large deviations from the original input. The overall actor loss $L_{\max} + \lambda_p L_{\text{prox}}$ balances validity with proximity.

Through repeated interactions with the black-box classifier and joint optimization of both actor and critic, the actor progressively learns a policy for generating counterfactuals that achieve the desired label change while remaining close to the original time series.

**Inference Workflow** Once the actor and supporting components have been trained, Algorithm 2 describes the inference procedure for generating counterfactual explanations on the testing dataset. The process begins by loading the pre-trained actor $\mu$ together with the encoder–decoder pair (line 1). The encoder maps the input instance $x$ into its latent representation $z$ (line 2), which provides a structured space in which the actor can operate. At the same time, the black-box model $M$ is queried to obtain the original prediction $y_M$ (line 3), which is used alongside the target label $y_T$ to guide counterfactual generation.

Given these inputs, the actor network produces a counterfactual latent embedding $z_{CF}$ (line 4). This representation is then decoded back into the input space using the decoder to yield the counterfactual sequence $x_{CF}$ (line 5). The final counterfactual instance is then returned (line 6). In practice, this inference workflow enables efficient and scalable generation of counterfactuals, since the trained actor produces explanations in a single forward pass without requiring per-instance optimization.

---

**Algorithm 1** Training procedure

---

**Inputs:** $M$: black-box model; proximity loss weight $\lambda_p$; total steps $T$; update interval $U$; batch size $|B|$; exploration std. $\sigma$; mixing coefficient $\beta=0.5$

**Outputs:** Trained actor network $\mu$ for counterfactual generation

**Note:** Each training step generates *one* counterfactual experience $(s, a, R)$ without sequential state transitions.

1: Load pre-trained encoder $enc$ and decoder $dec$
2: Initialize actor $\mu(\cdot; \theta_\mu)$ and critic $Q(\cdot; \theta_Q)$
3: Initialize replay buffer $\mathcal{D}$
4: Define reward function $f(\cdot, \cdot)$
5: **for** $t = 1$ to $T$ **do** ▷ $T$ training steps; each generates ONE counterfactual experience
6:     Sample input time series $x$ and target class $y_T$
7:     $y_M \leftarrow M(x); \quad z \leftarrow enc(x)$
8:     $z_{CF} \leftarrow \mu(z, y_M, y_T; \theta_\mu)$
9:     $\tilde{z}_{CF} \leftarrow \text{clip}(z_{CF} + \varepsilon, -1, 1), \quad \varepsilon \sim \mathcal{N}(0, 0.1)$
10:     $\tilde{x}_{CF} \leftarrow dec(\tilde{z}_{CF})$
11:     $R \leftarrow f(M(\tilde{x}_{CF}), y_T)$ ▷ Immediate binary reward: 1 if flip, 0 otherwise
12:     Store $(z, y_M, y_T, \tilde{z}_{CF}, R)$ in the replay buffer $\mathcal{D}$ ▷ No next state—episode ends
13:     **if** $t \bmod U = 0$ **then**
14:         Sample batch $\mathcal{B}$ of size $B$ from $\mathcal{D}$
15:         Update critic by one-step gradient descent using:

$$\nabla_{\theta_Q} \frac{1}{|B|} \sum_B \left( Q(z, y_M, y_T, \tilde{z}_{CF}) - R \right)^2$$

16:         Recompute $z_{CF} \leftarrow \mu(z, y_M, y_T; \theta_\mu), \ x_{CF} \leftarrow dec(z_{CF})$

$$L_{\max} = -\frac{1}{|B|} \sum_B Q(z, y_M, y_T, z_{CF})$$

$$L_{\text{prox}} = \frac{1}{|B|} \sum_B \left( \beta \|x - x_{CF}\|_1 + \|x - x_{CF}\|_2^2 \right) \quad \text{with } \beta = 0.5$$

17:         Update actor by one-step gradient descent using:

$$\nabla_{\theta_\mu} \left( L_{\max} + \lambda_p L_{\text{prox}} \right)$$

18:     **end if**
19: **end for**

---

**Algorithm 2** Generating explanations

---

**Inputs:** $x$: original instance; $y_T$: target label; $M$: black-box model

**Outputs:** $x_{CF}$: counterfactual instance

1: Load pre-trained actor $\mu$, encoder $enc$, and decoder $dec$
2: Compute latent representation $z \leftarrow enc(x)$
3: Obtain model prediction $y_M \leftarrow M(x)$
4: Generate counterfactual latent $z_{CF} \leftarrow \mu(z, y_M, y_T)$
5: Decode $x_{CF} \leftarrow dec(z_{CF})$
6: **return** $x_{CF}$ as the counterfactual instance

---

## 5 EXPERIMENTS

### 5.1 EXPERIMENTAL SETUP

**Datasets.** We evaluate our approach on a diverse collection of publicly available univariate time series datasets drawn from the University of California, Riverside (UCR) Time Series Classification Archive (Dau et al., 2019). The UCR archive is a widely used benchmark suite for time series classification and has been extensively adopted to assess both traditional and deep learning models. To ensure that our findings are representative across different domains and levels of complexity,

we select 13 datasets spanning application areas such as spectro, ECG, human activity recognition (HAR), sensor, traffic, etc. Table 1 in the Appendix provides detailed statistics for all datasets, including sequence length, number of classes, and train–test splits. These datasets have been shown to yield strong performance with state-of-the-art deep classifiers (Ismail Fawaz et al., 2019), making them suitable for evaluating the quality of counterfactual explanations.

The chosen datasets also vary in size, covering scenarios from small-scale problems with fewer than 100 training instances to medium-scale collections with 100–250 training samples, and to larger-scale datasets containing thousands of examples. This diversity allows us to systematically investigate how our method scales with training set size. Importantly, the datasets are partitioned following the standard UCR setup, where training and testing sets are evenly sampled across classes to avoid class imbalance issues.

**Baselines.** We compare our reinforcement learning framework against several state-of-the-art counterfactual explanation methods for time series classification. These baselines span optimization-based, saliency-guided, and latent-space perturbation approaches, representing the primary categories of existing techniques.

*Wachter.* Wachter et al. (2017) introduced one of the earliest general-purpose counterfactual frameworks. The method formulates counterfactual search as minimizing a weighted loss that balances prediction validity with input proximity, typically measured by the $L_1$ norm. Although originally designed for tabular data, it has been widely adapted as a baseline in time series settings (Delaney et al., 2021; Li et al., 2022a).

*TimeX.* Building on Wachter, Filali Boubrahimi & Hamdi (2022) enhances plausibility by incorporating Dynamic Barycenter Averaging (DBA) into the loss. This encourages counterfactuals to move toward the centroid of the target class under dynamic time warping (DTW), thereby improving interpretability and contiguity of perturbations.

*Info-CELS.* Li et al. (2025) extend the saliency-guided counterfactual explainer CELS (Li et al., 2023) by removing the thresholding step in saliency map generation. This adjustment eliminates noise from hard thresholds, producing smoother perturbations and significantly improving the validity of counterfactuals, while maintaining sparsity and proximity.

*Glacier (AE variants).* Wang et al. (2024) proposes a unified framework that performs gradient-based counterfactual search either in latent space or directly in input space, under different temporal constraints. In this study, we focus on the latent-space variants (`Glacier-AE`), where optimization is performed on autoencoder representations, balancing prediction-margin loss with constraint penalties to enforce sparsity and temporal plausibility. In particular, local constraints derived from LIMESegment (Sivill & Flach, 2022) highlight influential subsequences, guiding perturbations toward instance-specific regions.

**Black-box Classifiers.** For all counterfactual explanation methods, we employed the Fully Convolutional Network (FCN) as the black-box classifier. This provides a consistent evaluation framework across methods while supporting gradient-based approaches such as Wachter, TimeX, Info-CELS, and Glacier. In addition, the FCN has demonstrated strong and stable classification performance across diverse UCR datasets (Ismail Fawaz et al., 2019), making it a reliable backbone and ensuring that counterfactual evaluations are not affected by poor predictive accuracy. Further architectural and training details of the FCN are provided in the Appendix.

**Evaluation Metrics.** We evaluated the performances of different counterfactual models in terms of three major metrics:

*(1) Validity Metric (Flip Label Rate)*: This metric measures how often the generated counterfactuals successfully change the model's original prediction. Formally, it is defined as:

$$flip\_rate = \frac{num\_flipped}{num\_testsample}, \tag{8}$$

where *num_flipped* is the number of counterfactuals that flip the predicted label, and *num_testsample* is the total number of test inputs. A higher flip rate (closer to 1) indicates better validity, as more counterfactuals satisfy the label change requirement.

*(2) Proximity Metrics ($L_1$, $L_2$, and $L_\infty$).* Proximity evaluates how close each counterfactual is to the original input. We report three common distance-based metrics: (a) the $L_1$ Distance (Manhattan

distance) measures the total absolute change across all time steps; (b) the $L_2$ Distance (Euclidean distance): Emphasizes larger differences by squaring deviations; (c) the $L_\infty$ Distance (Chebyshev distance): Captures the maximum absolute deviation.

*(3) Plausibility Metrics. (IF, LOF, OC_SVM)* We evaluate plausibility by comparing the generated counterfactuals against the training dataset and computing how "out-of-distribution" they appear under three standard unsupervised anomaly detectors: (a) **Isolation Forest (IF)** (Liu et al., 2008): Detects anomalies by randomly partitioning data and isolating outliers with fewer splits; (b) **Local Outlier Factor (LOF)** (Breunig et al., 2000): Measures how much a sample deviates from its local neighborhood density; (c) **One-Class SVM (OC_SVM)** (Schölkopf et al., 2001): Learns the boundary of normal data to identify samples that fall outside the learned distribution. Each detector is trained only on the original training data, which defines the normal data manifold. We then feed the generated counterfactuals to these detectors and measure their outlier scores. Counterfactuals that receive lower outlier scores are considered more plausible, since they lie closer to the distribution of real time series in the training set.

## 5.2 Experimental Results

**Validity (Flip Rate):** Figure 3 compares the validity of counterfactuals across methods, measured by flip rate. The violin plot shows the distribution of flip rates across datasets. (*The black horizontal line and red dot represent the median and mean across datasets, respectively. This is kept consistent across all violin plots in the paper.*) Our RL framework consistently achieves strong performance, with validity concentrated in the 0.8–1.0 range and a high mean and median ( 0.95), reflecting reliable performance across datasets. This indicates that nearly all generated counterfactuals successfully flip the classifier's decision, a critical property for actionable explanations. TimeX and InfoCELS often reach near-perfect validity ($\approx$1.0), but their performance is less reliable, dropping below 0.7 on certain datasets. Glacier is even more fragile: while it can achieve high validity on some datasets, it collapses to zero on others. These fluctuations reduce the overall robustness of the baselines. By contrast, RL maintains a compact distribution, whereas the wider spread of InfoCELS, TimeX, and Glacier reveals susceptibility to dataset-specific failures. Taken together, these results highlight that RL provides a stable and generalizable mechanism for generating valid counterfactuals in diverse time series domains.

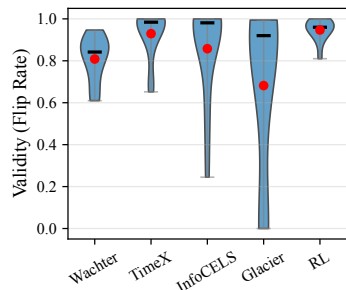

Figure 3: Validity evaluation measured by flip rate, where higher values indicate more valid counterfactuals.

**Proximity (L1, L2, $L_\infty$):** Figure 4 reports the proximity of generated counterfactuals, where lower values indicate smaller perturbations from the original inputs. For L1 and L2 distances, InfoCELS, TimeX, and Wachter achieve the lowest values. InfoCELS attains the best proximity by perturbing only the most salient time steps, while TimeX and Wachter maintain relatively low distances through explicit distance-based penalties in their optimization objectives. By contrast, RL and Glacier operate in latent space rather than directly perturbing the input, which leads to higher L1 and L2 values after decoding, as modifications are distributed more broadly across the sequence. Although RL employs an elastic distance penalty within the actor network to constrain perturbations, the decoding process through the autoencoder inevitably spreads changes across multiple time steps, yielding larger aggregate distances.

For $L_\infty$, which captures the largest pointwise deviation, RL ranks behind Wachter but demonstrates competitive performance with TimeX and InfoCELS, while clearly outperforming Glacier. This suggests that although RL produces broader perturbations—reflected in its higher $L_1$ and $L_2$ distances—it effectively avoids extreme spikes at individual time steps, leading to smoother and more controlled modifications. This behavior stems from the method's design: the autoencoder embedding promotes globally distributed adjustments rather than highly localized changes. As a result, RL yields counterfactuals that are less proximate in aggregate distance but remain better aligned with the underlying data manifold.

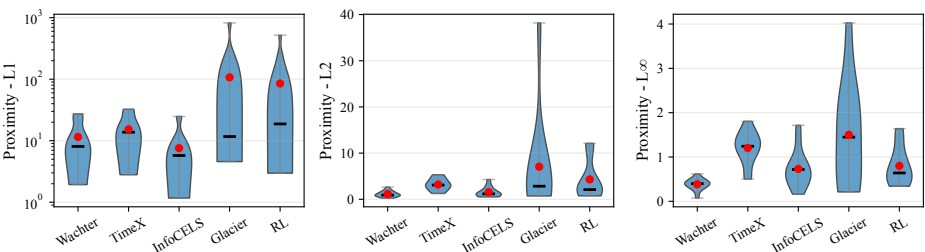

Figure 4: Proximity evaluation using three distance metrics. Lower values indicate counterfactuals closer to the original inputs.

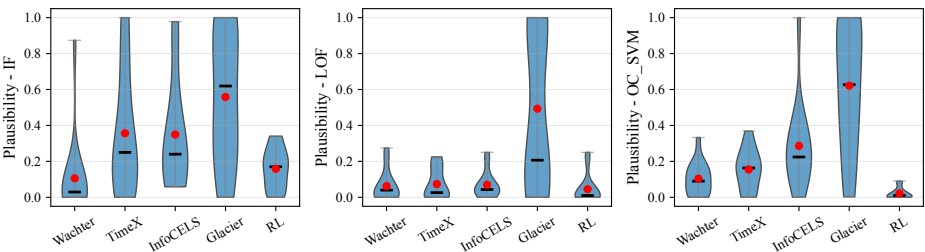

Figure 5: Plausibility evaluation using three outlier detection methods. Lower scores indicate more realistic counterfactuals.

**Plausibility (IF, LOF, OC_SVM):** Figure 5 shows that RL maintains consistently low outlier scores across all three detectors. Under IF, RL remains within the $0$–$0.35$ range, with both mean and median around $0.15$, while the baselines span the full $0$–$1$ interval with higher variance and frequent outliers. For LOF, RL stays below $0.3$ across all datasets, achieving the lowest mean and median values (close to $0$) and performing competitively with Wachter, TimeX, and InfoCELS, whereas Glacier again reaches up to $1.0$. Under OC_SVM, RL clearly outperforms all baselines, keeping scores consistently below $0.1$ with mean and median near zero, while Wachter and TimeX extend toward $0.4$ and Glacier fluctuates widely between $0$–$1$.

These results demonstrate that RL counterfactuals align more closely with the data manifold, benefiting from latent-space policy learning that encourages globally coherent perturbations rather than localized, unrealistic changes. By operating in the latent space, RL captures structural regularities of the data, which translates into counterfactuals that are not only valid but also plausible under multiple outlier detection metrics. In contrast, the baselines either compromise plausibility for proximity—achieving lower distances at the cost of more out-of-distribution samples—or show instability across datasets, leading to less reliable counterfactuals. Overall, the evidence highlights RL as a method that balances validity, proximity, and plausibility, producing counterfactuals that are both effective for model explanation and realistic within the data distribution.

## 6 CONCLUSION

In this paper, we developed an RL-based framework for counterfactual explanation in time series classification using a one-step actor–critic formulation. The agent learns a policy in the latent space of a pre-trained autoencoder to generate perturbations that flip the classifier's prediction, with the reward encouraging valid counterfactuals and a proximity penalty helping maintain realistic structure. Once trained, the actor generates counterfactuals in batches through a single forward pass, making the approach more scalable than instance-based optimization. Experimental results on 13 benchmark datasets show that our method consistently produces counterfactuals that are both valid and plausible. An important direction for future work is to extend our evaluation beyond the UCR collection to multivariate, irregular, or domain-specific datasets to further examine the generalizability of RL-based counterfactual generation.

REPRODUCIBILITY STATEMENT

To ensure reproducibility, we will release the full source code on an anonymous GitHub repository (https://github.com/bubbleblue0/Counterfactual-Explanations-for-Time-Series-Data-via-Reinforcement-Learning). Detailed descriptions of the datasets, hyperparameters, and model architectures (classifier, autoencoder, and actor–critic networks) are also provided in the Appendix, enabling researchers to replicate our experiments and results.

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

Table 1: UCR Datasets Metadata, Classifier Accuracy, and Autoencoder Best Validation Loss

| ID | Dataset Name | $\mathcal{C}$ | $\mathcal{L}$ | DS train size | DS test size | Type | FCN Acc. | AE Val Loss |
|----|--------------|----|-----|---------------|--------------|------|----------|-------------|
| 0 | Chinatown | 2 | 24 | 20 | 343 | TRAFFIC | 0.98 | 0.01 |
| 1 | Coffee | 2 | 286 | 28 | 28 | SPECTRO | 1.00 | 0.00 |
| 2 | ECG200 | 2 | 96 | 100 | 100 | ECG | 0.91 | 0.06 |
| 3 | FordA | 2 | 500 | 3601 | 1320 | SENSOR | 0.93 | 0.22 |
| 4 | FordB | 2 | 500 | 3636 | 810 | SENSOR | 0.81 | 0.15 |
| 5 | FreezerRegularTrain | 2 | 301 | 150 | 2850 | DEVICE | 1.00 | 0.01 |
| 6 | GunPoint | 2 | 150 | 50 | 150 | HAR | 1.00 | 0.01 |
| 7 | GunPointAgeSpan | 2 | 150 | 135 | 316 | HAR | 1.00 | 0.01 |
| 8 | GunPointMaleVersusFemale | 2 | 150 | 135 | 316 | HAR | 1.00 | 0.01 |
| 9 | GunPointOldVersusYoung | 2 | 150 | 135 | 316 | HAR | 0.99 | 0.01 |
| 10 | HandOutlines | 2 | 2709 | 1000 | 370 | IMAGE | 0.82 | 0.04 |
| 11 | TwoLeadECG | 2 | 82 | 23 | 1139 | ECG | 1.00 | 0.01 |
| 12 | Wafer | 2 | 152 | 1000 | 6164 | SENSOR | 1.00 | 0.01 |

$\mathcal{C}$: number of classes; $\mathcal{L}$: sequence length. FCN Acc. reports classifier accuracy on the test set. AE Val Loss corresponds to the lowest validation loss achieved during autoencoder training.

Zhendong Wang, Isak Samsten, Ioanna Miliou, Rami Mochaourab, and Panagiotis Papapetrou. Glacier: guided locally constrained counterfactual explanations for time series classification. *Machine Learning*, pp. 1–31, 2024.

Zhiguang Wang, Weizhong Yan, and Tim Oates. Time series classification from scratch with deep neural networks: A strong baseline. In *2017 International joint conference on neural networks (IJCNN)*, pp. 1578–1585. IEEE, 2017.

# A  APPENDIX

## A.1  LLM USAGE

GPT-5 was used as an assistive tool during the preparation of this manuscript. Specifically, GPT-5 was employed to help refine the writing style and improve the readability of our manuscript. All core research ideas, experiments, implementations, and analyses were conducted by the authors.

## A.2  THE DETAILS OF DATASETS USED IN OUR EXPERIMENTS

This section provides detailed metadata of the UCR datasets employed in our experiments. For each dataset, we report the number of classes, sequence length, training and testing sizes, domain type, and the classification accuracy of the black-box model.

## A.3  ABLATION STUDY

To evaluate the contribution of *experience replay* in our RL-based counterfactual generation framework, we conduct an ablation study comparing the full method (RL) with a variant that removes the replay buffer (RL w/o Replay). The goal of this ablation is to examine how replay affects the performance of generated counterfactuals across three key metrics: validity, proximity, and plausibility.

From Figure 6, we observe that experience replay contributes differently across the three evaluation metrics, with varying levels of impact on validity, proximity, and plausibility. Importantly, both proximity and plausibility are computed *only on valid counterfactuals*, ensuring that these results directly reflect the quality of CFs that successfully flip the label.

For validity, replay helps produce a tighter and higher flip-rate range of $[0.8, 1.0]$, whereas the no-replay variant has a wider range of $[0.5, 0.99]$. Although the median validity without replay is slightly higher and the mean slightly lower than with replay, this indicates that not all datasets require replay to maintain high validity of counterfactuals generated; some datasets could generate high validity counterfactuals even without it. For proximity, all three metrics ($L_1$, $L_2$, and $L_\infty$) show only minor differences between the replay and no-replay variants. The values are generally comparable across both settings, with no large or systematic shift. The replay variant exhibits slightly lower mean and median values across datasets, but the overall effect is modest. This indicates that replay

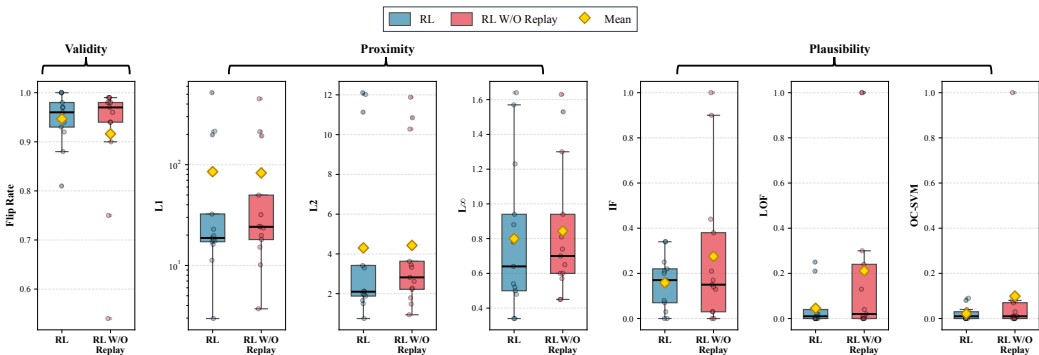

Figure 6: Ablation study comparing the full RL method with experience replay against the variant without replay. The figure reports the distribution, mean values, and median values (indicated by the bold black central horizontal line inside each boxplot) of validity, proximity (L1, L2, $L_\infty$), and plausibility (IF, LOF, OC-SVM) across all datasets.

does not strongly influence how close the counterfactuals remain to the original inputs. The largest performance difference appears in the plausibility metrics (IF, LOF, and OC-SVM). Even though the no-replay variant produces *valid* counterfactuals (i.e., they successfully flip the label), these valid CFs exhibit noticeably higher OOD scores and substantially larger variance, indicating that the perturbations are more likely to deviate from the data manifold. In comparison, the replay variant consistently produces lower and more stable plausibility values. These results collectively show that experience replay plays an important role in generating valid counterfactuals that are also plausible and manifold-consistent.

### A.4 THE ARCHITECTURE OF FCN MODEL

The FCN architecture follows Wang et al. (2017) and the implementation of Ismail Fawaz et al. (2019). It consists of three one-dimensional convolutional blocks: the first applies 128 filters of size 8, the second 256 filters of size 5, and the third 128 filters of size 3. Each convolution is followed by batch normalization and a ReLU activation. A global average pooling layer aggregates temporal features, and a final dense layer with softmax activation outputs class probabilities. Training is performed with the Adam optimizer (learning rate $10^{-3}$) and categorical cross-entropy loss. To improve convergence and prevent overfitting, we employ early stopping (patience 100, monitoring validation accuracy) and a learning rate scheduler that reduces the learning rate by a factor of 0.5 if the training loss plateaus for 30 epochs. Models are trained for up to 2000 epochs with a batch size of 16. For smaller datasets, we use a reduced mini-batch size of $\min(N/10, 16)$, where $N$ denotes the number of training instances, to ensure stable optimization. The classification accuracy of the trained FCN on each test dataset is reported in Table 1.

### A.5 THE ARCHITECTURE OF AUTOENCODER

For counterfactual generation, we employ a convolutional autoencoder that provides a structured latent space for perturbations. The encoder consists of two one-dimensional convolutional layers with ReLU activations, each followed by max-pooling to progressively downsample the input. A fully connected layer with a `tanh` activation then maps the extracted features into a latent representation bounded within $[-1, 1]$. We adopt this bounded latent range for two reasons: (1) it matches the range observed during decoder training, ensuring that latent vectors remain within the in-distribution region of the autoencoder; and (2) it aligns with standard practice in actor–critic methods where keeping actions within a compact set stabilizes optimization and avoids exploding activations or gradients. In addition to the `tanh` activation, we also apply hard clipping to the latent actions during counterfactual generation. Although the encoder outputs are naturally bounded by `tanh`, exploration noise and actor updates may push latent vectors outside the $[-1, 1]$ interval. Such out-of-range values can lead to out-of-distribution inputs for the decoder and degrade reconstruction fidelity. Clipping prevents these issues by ensuring that both the actor outputs and

the replayed latent actions remain within the decoder's training range, thereby improving numerical stability and maintaining coherent reconstructions. The decoder reconstructs time series from this latent space by first expanding the latent code with a fully connected layer reshaped into a sequence, followed by causal one-dimensional convolutions and custom upsampling layers that preserve temporal ordering; a final convolutional layer with linear activation outputs a sequence with the original number of features, with cropping or last-value padding applied as needed to match the exact input length. The autoencoder was trained separately for each dataset using mean squared error (MSE) reconstruction loss and the Adam optimizer with a learning rate of 0.005. Training was performed for up to 2000 epochs with a batch size of 128, reduced to $\min(N/10, 128)$ for small datasets where $N$ is the number of training samples, and employed early stopping with patience 50 together with learning rate reduction on plateau (factor 0.5, patience 30, minimum learning rate $10^{-6}$). During training, the model was monitored using validation loss, and the weights with the lowest validation loss were saved. The best validation reconstruction losses obtained from these saved models are reported in Table 1.

**Impact of Autoencoder Performance on Counterfactual Quality.** Our RL agent operates entirely in the latent space of a pre-trained autoencoder. Therefore, the reconstruction quality of this autoencoder has a direct effect on counterfactual generation. A well-trained autoencoder provides a reliable latent manifold and ensures that decoded perturbations remain coherent and realistic. Specifically: (1) **Meaningful latent manifold.** High-quality reconstructions indicate that the encoder captures the dominant temporal structure of the dataset. The RL policy relies on this structure when generating perturbed latent vectors. If the latent space is poorly organized, even small perturbations may decode into unrealistic or noisy sequences. (2) **Stable proximity optimization.** The actor is optimized using the proximity term (see equation 6), which depends directly on the decoder output. When reconstruction error is high, the proximity loss becomes unreliable, leading to unstable or misleading gradient signals during actor updates. (3) **Plausibility of decoded counterfactuals.** Since plausibility metrics (IF, LOF, and OC-SVM) evaluate counterfactuals in the input space, the decoder's ability to reconstruct realistic time series is crucial. A poorly trained autoencoder may map valid latent perturbations to out-of-distribution sequences, inflating outlier scores.

## A.6 ACTOR–CRITIC ARCHITECTURES

The actor and critic networks used in our framework follow the standard design commonly adopted in deep reinforcement learning. Both models employ fully connected layers with layer normalization and ReLU activations, which provide stable training dynamics across tasks.

Specifically, the actor maps the state representation into a continuous action through a three-layer multilayer perceptron (MLP). Two hidden layers (each of size 256 with ReLU activations) are followed by an output layer with a $\tanh$ activation, which ensures bounded perturbations in the latent space. This network is responsible for generating candidate counterfactual representations. And the critic estimates the value of a state–action pair using a similar three-layer MLP. Two hidden layers of size 256 (with ReLU activations) are followed by a scalar output layer that predicts the $Q$-value. The critic provides feedback to guide the actor's updates by evaluating the expected reward of the generated counterfactuals.

## A.7 IMPLEMENTATION DETAILS OF REINFORCEMENT LEARNING

For counterfactual generation, we adopt the reinforcement learning framework implemented in the `alibi` library, using a Deep Deterministic Policy Gradient (DDPG) setup. The CFRL agent consists of an actor network, which generates perturbations, and a critic network, which estimates their quality. Both networks are trained jointly with additional sparsity and consistency losses to guide the learning process.

The training procedure is run for 50,000 steps, with dataset-specific batch sizes ranging from 16 to 256 depending on the dataset size (see code for mapping). The actor loss combines the policy gradient term (negative critic output) with proximity penalties, weighted by coefficients $\lambda_{\text{prox}} = 1$. The critic is trained by minimizing the squared error between its predicted $Q$-values and the observed rewards. To ensure stable training, Gaussian noise is added to the actor's output for exploration.

Table 2: Validity evaluation (Flip Rate) across UCR datasets. Higher values indicate more valid counterfactuals.

| ID | Dataset | Wachter | TimeX | InfoCELS | Glacier | RL |
|----|---------|---------|-------|----------|---------|-----|
| 0 | Chinatown | 0.8921 | 1.0000 | 0.2449 | 0.9038 | 0.9600 |
| 1 | Coffee | 0.7857 | 1.0000 | 1.0000 | 0.9286 | 1.0000 |
| 2 | ECG200 | 0.8800 | 0.9800 | 1.0000 | 0.9200 | 0.9700 |
| 3 | FordA | 0.8674 | 0.9985 | 0.4955 | 0.6432 | 0.9400 |
| 4 | FordB | 0.6099 | 0.9222 | 0.9889 | 0.9444 | 0.8100 |
| 5 | FreezerRegularTrain | 0.6898 | 0.6519 | 0.9811 | 0.9825 | 1.0000 |
| 6 | GunPoint | 0.9467 | 1.0000 | 1.0000 | 0.9667 | 0.9200 |
| 7 | GunPointAgeSpan | 0.6171 | 0.8734 | 0.6582 | 0.0000 | 0.8800 |
| 8 | GunPointMaleVersusFemale | 0.8418 | 0.9968 | 0.9367 | 0.0000 | 0.9500 |
| 9 | GunPointOldVersusYoung | 0.8317 | 0.9841 | 0.8984 | 0.0000 | 0.9700 |
| 10 | HandOutlines | 0.7676 | 0.7108 | 0.9919 | 0.9946 | 1.0000 |
| 11 | TwoLeadECG | 0.8946 | 1.0000 | 1.0000 | 0.6119 | 0.9300 |
| 12 | Wafer | 0.8845 | 0.9655 | 0.9550 | 0.9685 | 0.9800 |

The predictor (black-box classifier) is kept fixed and only used to compute flip-label rewards. The autoencoder encoder–decoder pair is also fixed during RL training, providing the latent space for perturbations and reconstruction of counterfactuals. Rewards, success rates, and loss terms are monitored throughout training using Weights & Biases (wandb), and additional callbacks log intermediate samples and visualizations every 100 steps.

At the end of training, the learned actor network can generate counterfactuals in batches through a single forward pass, while the critic provides a learned notion of counterfactual quality. All reported results are based on the trained actor, without further fine-tuning at inference time.

### A.8 THE FULL TABLE FOR OUR EXPERIMENTAL RESULTS

In the main content, we visualize the overall performance of different methods using violin plots that aggregate results across all datasets. Tables 2-8 in the Appendix provide the full tabular results for each dataset. These results complement the violin plots in the main content by offering detailed, dataset-specific performance metrics. The reported metrics include **Validity** (measured by flip rate), **Proximity** (average $L_1$, $L_2$, and $L_\infty$ distances), and **Plausibility** (measured by outlier scores from Isolation Forest (IF), Local Outlier Factor (LOF), and One-Class SVM (OC_SVM)). For proximity and plausibility, the averages are computed only over valid counterfactuals for each dataset, consistent with the procedure used in the violin plots, ensuring that evaluation reflects the quality of successful counterfactuals.

Among all of the datasets, FordB stands out as one of the most challenging cases. As reported in Table 1, the base classifier achieves the lowest test accuracy on FordB, indicating that the underlying decision boundary is more ambiguous and that many instances lie close to class boundaries. In such a setting, counterfactual generation is inherently more difficult: even small perturbations can move samples across unstable regions of the decision surface, and the model's predictions are less reliable as a target signal for the RL agent. At the same time, the autoencoder trained on FordB exhibits relatively higher validation reconstruction loss compared to other datasets (Table 1), suggesting that it is harder to learn a compact and smooth latent representation for this dataset. Since our method perturbs in the latent space and decodes back to the input domain, a less accurate autoencoder makes it more challenging to produce perturbations that are both close to the original sequence and effective in flipping the label. This combination of lower classifier accuracy and higher reconstruction error helps explain why the validity scores on FordB are lower than on other datasets in Table 2.

### A.9 EFFICIENCY ANALYSIS AND RUNNING-TIME COMPARISON

To compare the computational efficiency of different counterfactual generation approaches, we analyze the runtime of our RL-based framework relative to mainstream instance-based optimization

Table 3: Proximity evaluation measured by $L_1$ distance (mean values). Lower values indicate smaller perturbations from the original inputs.

| ID | Dataset | Wachter | TimeX | InfoCELS | Glacier | RL |
|---|---|---|---|---|---|---|
| 0 | Chinatown | 1.94 | 2.80 | 1.17 | 12.89 | 2.97 |
| 1 | Coffee | 17.08 | 6.18 | 6.10 | 10.17 | 18.74 |
| 2 | ECG200 | 7.54 | 6.30 | 8.92 | 10.15 | 22.89 |
| 3 | FordA | 27.25 | 27.16 | 24.79 | 28.67 | 214.94 |
| 4 | FordB | 8.07 | 14.87 | 24.48 | 129.26 | 197.88 |
| 5 | FreezerRegularTrain | 23.54 | 17.56 | 3.40 | 10.49 | 17.20 |
| 6 | GunPoint | 6.61 | 12.65 | 6.91 | 35.12 | 17.49 |
| 7 | GunPointAgeSpan | 8.50 | 18.88 | 2.53 | - | 19.74 |
| 8 | GunPointMaleVersusFemale | 24.88 | 32.46 | 5.96 | - | 32.41 |
| 9 | GunPointOldVersusYoung | 6.10 | 10.13 | 1.93 | - | 17.97 |
| 10 | HandOutlines | 8.97 | 32.05 | 5.75 | 819.71 | 518.62 |
| 11 | TwoLeadECG | 4.11 | 4.42 | 4.09 | 9.63 | 11.25 |
| 12 | Wafer | 5.07 | 13.75 | 2.37 | 4.58 | 16.13 |

Table 4: Proximity evaluation measured by $L_2$ distance (mean values). Lower values indicate smaller perturbations from the original inputs.

| ID | Dataset | Wachter | TimeX | InfoCELS | Glacier | RL |
|---|---|---|---|---|---|---|
| 0 | Chinatown | 0.66 | 1.46 | 0.52 | 4.26 | 0.75 |
| 1 | Coffee | 1.51 | 1.27 | 1.12 | 1.04 | 1.50 |
| 2 | ECG200 | 1.20 | 2.50 | 2.50 | 2.99 | 3.30 |
| 3 | FordA | 1.68 | 5.10 | 2.92 | 2.68 | 12.02 |
| 4 | FordB | 0.60 | 3.27 | 4.29 | 10.76 | 11.13 |
| 5 | FreezerRegularTrain | 1.90 | 4.25 | 1.26 | 1.19 | 1.99 |
| 6 | GunPoint | 0.91 | 3.02 | 1.65 | 6.36 | 1.88 |
| 7 | GunPointAgeSpan | 1.18 | 4.29 | 0.83 | - | 2.10 |
| 8 | GunPointMaleVersusFemale | 2.63 | 5.32 | 1.48 | - | 3.42 |
| 9 | GunPointOldVersusYoung | 0.79 | 2.58 | 0.67 | - | 1.96 |
| 10 | HandOutlines | 0.27 | 3.09 | 0.59 | 38.17 | 12.11 |
| 11 | TwoLeadECG | 0.84 | 1.95 | 1.18 | 2.33 | 1.66 |
| 12 | Wafer | 0.77 | 3.49 | 1.07 | 0.73 | 2.13 |

baseline methods. A key distinction is that baseline methods do not include a separate "training" phase for counterfactual generation—they optimize each test instance independently—whereas our RL approach consists of both a training stage (policy learning) and a fast inference stage (single forward pass). To provide a fair comparison, we report:

- **Training time** (RL only),

- **Inference time** (all methods), and

- **Total end-to-end time** (RL training + RL inference vs. inference-only baselines).

Figure 7 presents the runtime comparison across datasets of increasing size (ordered from smallest to largest). Sorting datasets by size makes it easier to observe how methods scale as the number of test samples grows. Because all instance-based methods require iterative optimization for every input, their runtime increases roughly linearly with dataset size. In contrast, the RL method is trained once on the dataset, and its training cost is independent of the number of test instances. During inference, each counterfactual is produced by a single actor forward pass, which takes only milliseconds, so the overall inference time grows linearly but very slowly with the size of the test set.

Table 5: Proximity evaluation measured by $L_\infty$ distance (mean values). Lower values indicate smaller maximum deviations from the original inputs.

| ID | Dataset | Wachter | TimeX | InfoCELS | Glacier | RL |
|---|---|---|---|---|---|---|
| 0 | Chinatown | 0.42 | 1.02 | 0.37 | 2.22 | 0.34 |
| 1 | Coffee | 0.40 | 0.50 | 0.45 | 0.21 | 0.34 |
| 2 | ECG200 | 0.47 | 1.40 | 1.39 | 1.80 | 1.23 |
| 3 | FordA | 0.36 | 1.80 | 0.86 | 0.63 | 1.64 |
| 4 | FordB | 0.25 | 1.24 | 1.71 | 2.30 | 1.57 |
| 5 | FreezerRegularTrain | 0.47 | 1.54 | 0.84 | 0.37 | 0.88 |
| 6 | GunPoint | 0.37 | 1.17 | 0.73 | 2.06 | 0.48 |
| 7 | GunPointAgeSpan | 0.47 | 1.48 | 0.50 | - | 0.50 |
| 8 | GunPointMaleVersusFemale | 0.62 | 1.45 | 0.75 | - | 0.78 |
| 9 | GunPointOldVersusYoung | 0.32 | 0.99 | 0.42 | - | 0.52 |
| 10 | HandOutlines | 0.07 | 0.52 | 0.16 | 4.02 | 0.54 |
| 11 | TwoLeadECG | 0.42 | 1.26 | 0.61 | 1.09 | 0.64 |
| 12 | Wafer | 0.34 | 1.24 | 0.72 | 0.26 | 0.94 |

Table 6: Plausibility evaluation measured by Isolation Forest (IF) outlier scores. Lower values indicate counterfactuals closer to the data manifold.

| ID | Dataset | Wachter | TimeX | InfoCELS | Glacier | RL |
|---|---|---|---|---|---|---|
| 0 | Chinatown | 0.00 | 0.00 | 0.60 | 1.00 | 0.34 |
| 1 | Coffee | 0.04 | 0.07 | 0.07 | 0.78 | 0.07 |
| 2 | ECG200 | 0.07 | 0.25 | 0.39 | 0.33 | 0.03 |
| 3 | FordA | 0.00 | 1.00 | 0.94 | 0.00 | 0.00 |
| 4 | FordB | 0.87 | 1.00 | 0.98 | 0.00 | 0.00 |
| 5 | FreezerRegularTrain | 0.00 | 0.63 | 0.24 | 0.22 | 0.21 |
| 6 | GunPoint | 0.10 | 0.30 | 0.36 | 0.62 | 0.22 |
| 7 | GunPointAgeSpan | 0.06 | 0.21 | 0.12 | - | 0.17 |
| 8 | GunPointMaleVersusFemale | 0.01 | 0.08 | 0.22 | - | 0.20 |
| 9 | GunPointOldVersusYoung | 0.03 | 0.14 | 0.06 | - | 0.15 |
| 10 | HandOutlines | 0.17 | 0.30 | 0.15 | 0.34 | 0.25 |
| 11 | TwoLeadECG | 0.00 | 0.00 | 0.09 | 0.65 | 0.08 |
| 12 | Wafer | 0.02 | 0.65 | 0.33 | 0.32 | 0.34 |

**Small Datasets.** On small datasets (e.g., `Coffee`, `ECG200`,`GunPoint`), instance-based methods such as Wachter, TimeX, InfoCELS, and Glacier have low total runtime simply because there are very few test samples. In these cases, the one-time training cost of the RL method becomes the main part of the total runtime, so the baselines may look faster. This is expected, since the benefit of training an RL policy only shows up when the test set is larger.

**Medium-Sized Datasets.** As we move to medium-sized datasets, the difference becomes more pronounced. Instance-based methods require optimization loops for an increasing number of samples, which causes their inference time to grow rapidly. In contrast, the RL method is trained once and its training cost stays within a reasonable range for all datasets. During inference, RL generates counterfactuals with a single forward pass, so inference time only grows linearly in the number of test instances and stays within seconds. As a result, the total runtime of the RL method becomes competitive with—and often lower than—the baseline methods on medium-sized datasets.

**Large Datasets.** For the largest datasets (e.g., `FordA`, `Wafer`, `FreezerRegularTrain`), the scalability benefits of RL becomes most significant. Baseline methods must run thousands of optimization loops, causing their runtime to increase sharply. Wachter, TimeX, InfoCELS, and Glacier

Table 7: Plausibility evaluation measured by Local Outlier Factor (LOF) scores. Lower values indicate counterfactuals closer to the data manifold.

| ID | Dataset | Wachter | TimeX | InfoCELS | Glacier | RL |
|---|---|---|---|---|---|---|
| 0 | Chinatown | 0.00 | 0.05 | 0.25 | 0.99 | 0.00 |
| 1 | Coffee | 0.00 | 0.14 | 0.04 | 0.08 | 0.00 |
| 2 | ECG200 | 0.00 | 0.00 | 0.03 | 0.21 | 0.00 |
| 3 | FordA | 0.00 | 0.00 | 0.00 | 0.00 | 0.00 |
| 4 | FordB | 0.00 | 0.00 | 0.00 | 0.00 | 0.00 |
| 5 | FreezerRegularTrain | 0.04 | 0.20 | 0.09 | 0.20 | 0.04 |
| 6 | GunPoint | 0.06 | 0.10 | 0.06 | 0.68 | 0.02 |
| 7 | GunPointAgeSpan | 0.06 | 0.02 | 0.10 | - | 0.02 |
| 8 | GunPointMaleVersusFemale | 0.27 | 0.19 | 0.04 | - | 0.21 |
| 9 | GunPointOldVersusYoung | 0.11 | 0.00 | 0.03 | - | 0.01 |
| 10 | HandOutlines | 0.22 | 0.23 | 0.20 | 0.99 | 0.25 |
| 11 | TwoLeadECG | 0.00 | 0.00 | 0.04 | 0.18 | 0.00 |
| 12 | Wafer | 0.05 | 0.03 | 0.04 | 0.10 | 0.04 |

Table 8: Plausibility evaluation measured by OC_SVM outlier scores. Lower values indicate counterfactuals closer to the data manifold.

| ID | Dataset | Wachter | TimeX | InfoCELS | Glacier | RL |
|---|---|---|---|---|---|---|
| 0 | Chinatown | 0.09 | 0.01 | 1.00 | 1.00 | 0.01 |
| 1 | Coffee | 0.00 | 0.29 | 0.35 | 0.23 | 0.00 |
| 2 | ECG200 | 0.07 | 0.20 | 0.16 | 0.52 | 0.01 |
| 3 | FordA | 0.00 | 0.00 | 0.38 | 0.52 | 0.09 |
| 4 | FordB | 0.00 | 0.00 | 0.00 | 0.00 | 0.08 |
| 5 | FreezerRegularTrain | 0.03 | 0.27 | 0.12 | 0.41 | 0.01 |
| 6 | GunPoint | 0.17 | 0.18 | 0.35 | 0.76 | 0.03 |
| 7 | GunPointAgeSpan | 0.13 | 0.16 | 0.40 | - | 0.00 |
| 8 | GunPointMaleVersusFemale | 0.16 | 0.12 | 0.16 | - | 0.00 |
| 9 | GunPointOldVersusYoung | 0.14 | 0.15 | 0.13 | - | 0.01 |
| 10 | HandOutlines | 0.33 | 0.37 | 0.22 | 0.92 | 0.00 |
| 11 | TwoLeadECG | 0.04 | 0.07 | 0.18 | 0.63 | 0.00 |
| 12 | Wafer | 0.18 | 0.19 | 0.26 | 0.09 | 0.04 |

all show substantial slowdowns because each counterfactual requires repeated gradient-based updates on every instance. In contrast, the RL method introduces no per-instance optimization: once trained, generating a counterfactual only requires an actor forward pass followed by decoding. Although RL inference still grows linearly with the number of test samples, each forward pass takes only milliseconds, making the overall inference cost very small. As a result, the RL method achieves the lowest total runtime among all approaches on large datasets. These results highlight that RL is especially advantageous for large-scale counterfactual generation scenarios.

**Additional Summary of Efficiency.** Beyond the trends shown in Figure 7, the runtime differences among baseline methods follow from how each method performs optimization. Glacier is usually faster than other instance-based methods because it optimizes in the autoencoder latent space, which is much lower-dimensional than the raw time series, making each gradient update cheaper. InfoCELS is faster than TimeX and ALIBI because it optimizes only a lightweight saliency mask, whereas TimeX and ALIBI repeatedly update full time-series inputs. Our RL method also operates in the latent space, but unlike Glacier, it does not re-optimize for every instance. Once the policy is trained, each counterfactual is generated with a single forward pass, which removes per-instance optimization entirely. This difference becomes increasingly important on large datasets, making RL the most scalable method among all approaches. It is also worth noting that a few datasets—such

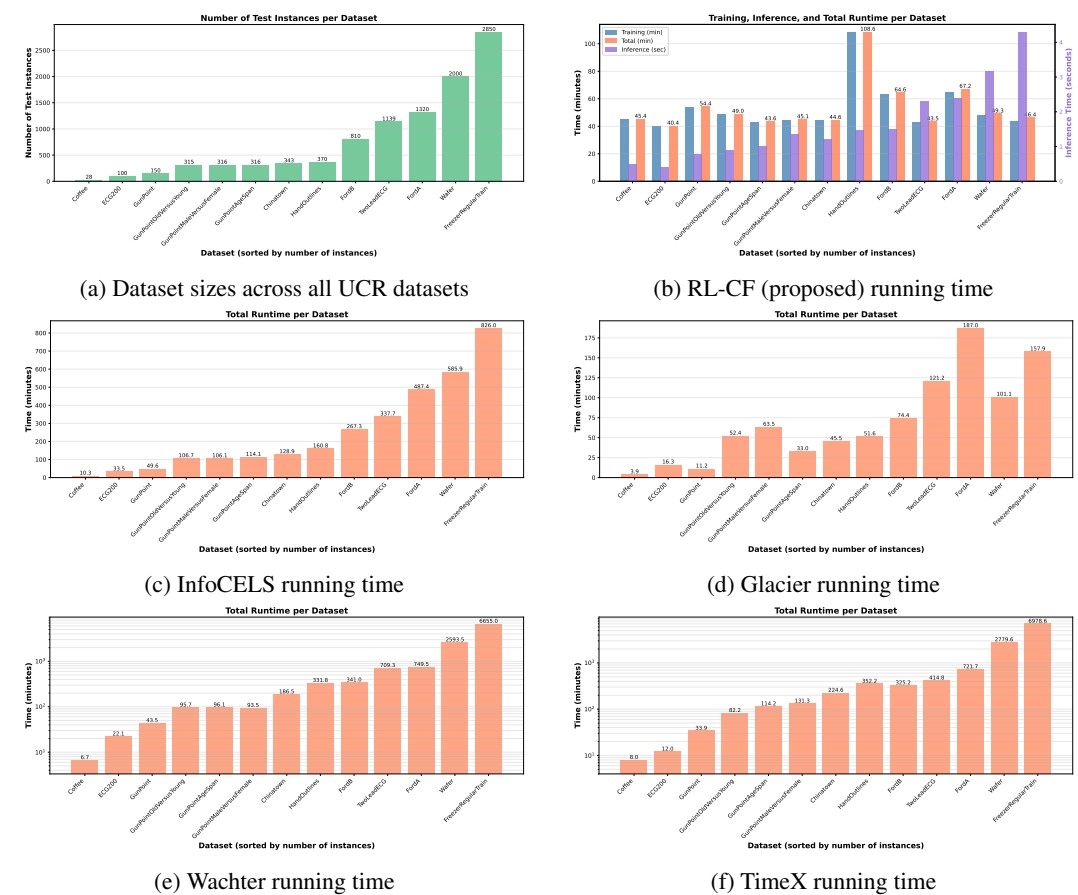

Figure 7: Running-time comparison of the proposed RL-CF method and baseline counterfactual explanation methods across all UCR datasets. Subfigure (a) shows the dataset sizes. Subfigure (b) presents the running time of our RL-CF method. Subfigures (c)–(f) report the runtime of InfoCELS, Glacier, Wachter, and TimeX, respectively. All results were obtained using the same hardware environment.

as HandOutlines, FordA, and FordB—show longer RL runtimes than others. These datasets either contain very long sequences (e.g., HandOutlines with length 2709) or have exceptionally large training sets (e.g., FordA and FordB with more than 3600 training samples). Longer sequences lead to higher-dimensional latent representations, which increase the computational cost of each actor and critic update, while larger training sets naturally require more training steps to learn an effective policy. Even so, the RL method still avoids per-instance optimization and remains substantially faster than the baselines on these datasets.

Overall, the runtime analysis in Figure 7 shows that instance-based counterfactual methods scale poorly as dataset size grows, since they must re-run iterative optimization for every test instance. In contrast, the RL approach trains once and then provides extremely fast inference—typically within seconds even for large test sets—because counterfactuals are produced by a single forward pass. As the number of test samples increases, this difference becomes more significant, and RL becomes the most efficient method among all baselines. These results demonstrate that experience-driven policy learning offers a scalable and reusable mechanism for counterfactual generation, especially in large-scale settings. This scalability advantage is particularly important for practical deployment scenarios in which models must support large test sets or high-volume counterfactual generation.

## A.10 Case Study: Qualitative Comparison of Counterfactual Explanations

To complement the quantitative evaluation, we conduct qualitative case studies on two representative UCR datasets—Coffee and GunPoint—to visually assess the temporal coherence and structural plausibility of counterfactuals generated by different methods. For each dataset, we plot all valid counterfactuals under each baseline and compare them with our proposed RL-generated counterfactuals. The corresponding visualizations are shown in Figures for Coffee (Figure 8a) and GunPoint (Figure 8b).

**Coffee.** For the Coffee dataset, the time series patterns within each class are highly consistent and smooth. Most baselines (TimeX, InfoCELS, Glacier) and our proposed method, RL, can maintain this temporal coherence when producing valid counterfactuals. The main exception is Wachter, whose pointwise optimization introduces noticeable fluctuations and local noise that depart from the characteristic smooth class structure.

**GunPoint.** In GunPoint, even instances from the same class exhibit substantial variation in magnitude and slope, making counterfactual generation more challenging. This intra-class diversity leads nearly all baselines (Wachter, TimeX, InfoCELS, Glacier) to produce counterfactuals with abrupt, irregular distortions and unstable peak shapes. The RL method is the only approach that consistently maintains the global rise–fall motion pattern, generating smooth and coherent counterfactuals despite the dataset's inherent variability.

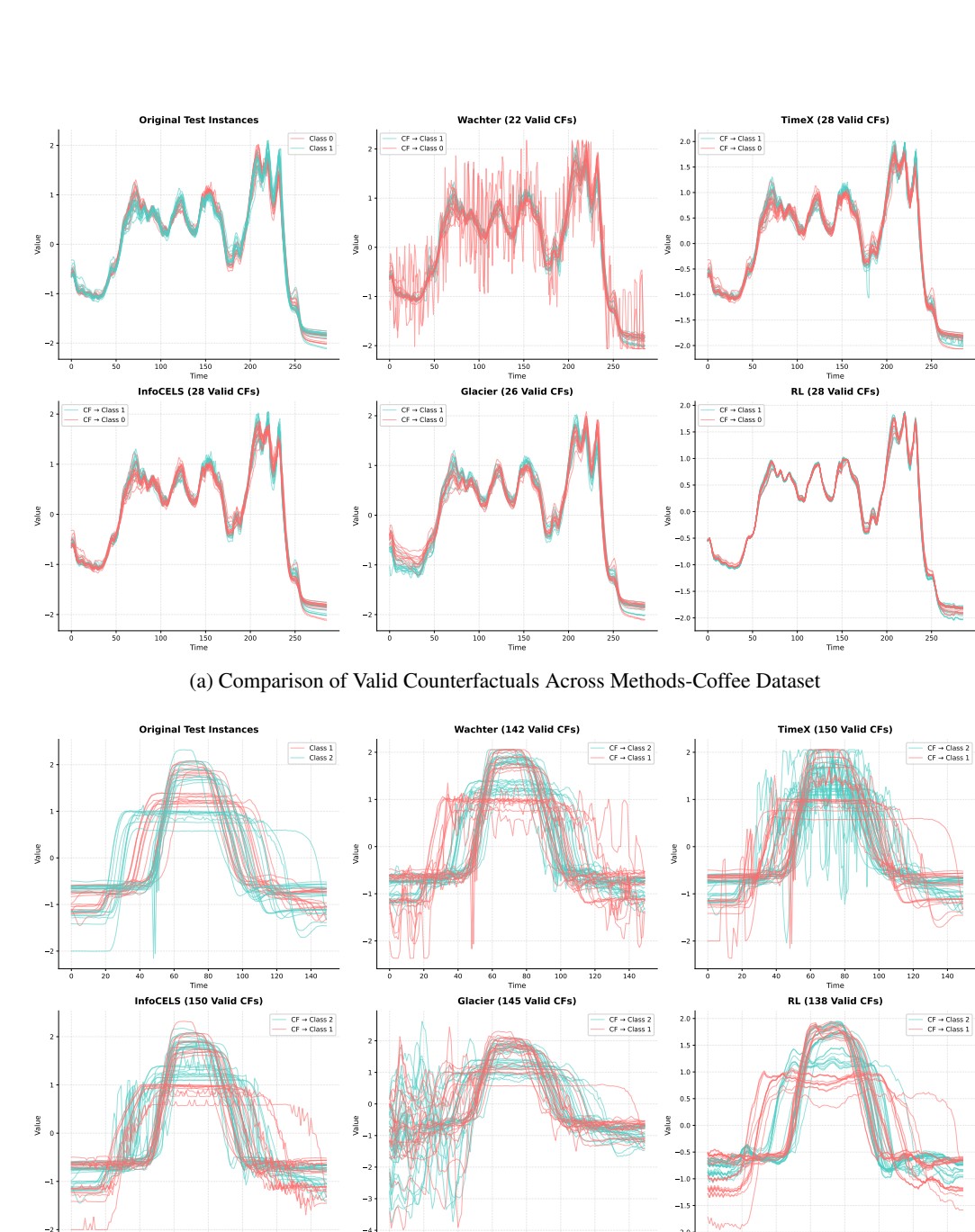

(a) Comparison of Valid Counterfactuals Across Methods-Coffee Dataset

(b) Comparison of Valid Counterfactuals Across Methods-Gunpoint Dataset

Figure 8: Qualitative Case Studies on Coffee and GunPoint Datasets

