# OpenReview forum: "Counterfactual Explanations for Time Series Data via Reinforcement Learning"
_ICLR.cc/2026/Conference — Submitted to ICLR 2026_

### Official Review · Reviewer_4v39 · 2025-10-29

**Soundness:** 2
**Presentation:** 3
**Contribution:** 2
**Rating:** 4
**Confidence:** 3

**Summary:**

The paper proposes a reinforcement-learning framework for generating CF explanations for time-series classifiers. A actor–critic agent (DDPG style) perturbs an autoencoder’s latent representation to produce CFs that flip a black-box model’s prediction while staying close to the original instance by an L1&L2 proximity penalty. Inference uses a single forward pass through the trained actor and decoder. A notable contribution is efficient batch CF generation at inference (Algorithm 2), avoiding per-instance optimization. Experiments on 13 UCR benchmark datasets compare the RL approach against several baselines. The results show high validity and strong plausibility (Fig. 1 and Fig. 3).

**Strengths:**

1. The paper targets a practical challenge: producing CF explanations for time-series classifiers where optimization-based methods can be slow and difficult to ensure plausibility. The proposed RL framework shifts most computation to training from testing, therefore at inference, the actor can generate CFs in a single forward pass and supports batching.

2. The method is model-agnostic, requiring only access to the classifier’s predictions for reward computation, without access of gradient to the black-box model.

3. Empirically, the approach consistently flips predictions across 13 benchmark datasets and shows high plausibility and robust validity relative to baselines.

4. The manuscript is clearly structured and easy to follow.

**Weaknesses:**

### 1. (Major) Clarity on the RL framing and one-step formulation.
While the paper frames CF generation as a sequential decision process in Section 1, it later states that the setup is “equivalent to a Markov decision process with a one-step horizon,” which removes sequential structure. This raises questions about the motivation for a DDPG-style actor–critic framework in this setting. Specifically:

a) The setup is explicitly defined as a MDP with a one-step horizon, this removes the sequential aspect of the search, and it seems to me, is more resembling a contextual bandit rather than a full RL problem.

b) The critic loss is defined as the squared error between the critic's estimate and the immediate reward, so it slightly deviates from standard DDPG and in this one-step setting, the critic acts merely as a reward predictor rather than the estimator of a long-term Q.

c) The benefits of experience replay and the complex, full actor–critic model over a simpler direct mapping/generative alternative are not clearly demonstrated. If such complexity is required by the black-box setting (e.g., the reward function can be non-differentiable for a black-box model), please explicitly state such motivations in the paper.

### 2. (Major) Trade-off in Proximity and Minimality:
Compared to optimization-based baselines, the method tends to produce CFs that are farther from the original inputs under L1/L2 distances, which may be conflicting with the “minimal change” goal. Larger distances can also reduce actionability in some application settings.

### 3. (Major) Lack of Ablation Studies and Qualitative Analysis:
a) Most empirical results are presented as violin plots, additional per-dataset tables or simple statistical comparisons would strengthen the claims. Also, key design choices (e.g., experience replay, clipping) are not ablated, making it hard to evaluate their contributions.

b) The paper does not report training cost (for Algorithm 1) or compare inference latency (generation of CFs) against optimization-based baselines to validate their advantages in batching.

c) Providing visualizable case studies of the generated counterfactuals will be helpful to evaluate the qualitative performance.

### 4. (Minor) Table 1 in Appendix overflows the right margin.

**If any of these points reflect a misunderstanding on my part, clarification is welcome.**

**Questions:**

1. Did the authors evaluate a simpler direct mapping (e.g., a single actor network trained to produce $z_{CF}$ without a critic), and if so, how is the outcome?

2. It seems that the text defines the action as a perturbation applied to the state, but Algorithm 1 indicates the policy outputs a latent CF $z_{CF}$ (with added noise and clipping) rather than a separate $\delta$. Could the authors clarify this definition?

3. What is the motivation for clipping latents to [−1,1]?

---

> ### Author Response · Authors · 2025-11-23
> **Response to Reviewer 4v39**
>
> **W1**: The reviewer is correct. Our implementation is a one-step MDP where each counterfactual generation is a single decision (no sequential states). We sincerely thank the reviewer for these insightful comments. They helped us identify places where our framing could be misunderstood. In the revised manuscript, we have clarified the RL formulation and corrected earlier wording that unintentionally suggested a fully sequential MDP. More concretely, we now explicitly state that the problem is formulated as a contextual bandit (i.e., a one-step MDP) rather than a sequential decision process. The misleading phrasing in Section 1 has been removed, and the formal description in Section 4 (**page 4, starting from line 214**) has been revised to accurately reflect the one-step setting. We also clarify why an actor–critic structure remains useful in this context—specifically, the non-differentiable reward signal, the need for a stable function approximator of expected reward, and the benefits of experience replay even in single-step problems. The role of the critic as an immediate-reward estimator (rather than a long-horizon Q-function) is now explicitly stated. These revisions appear in Section 1 and Section 4 of the paper, and we appreciate the reviewer’s feedback, which helped improve the clarity and correctness of our methodological presentation.
>
> **W2:** Thank you for this excellent observation. We agree that our method does not achieve the smallest L1 or L2 distances. As described in Equation (6), we explicitly include a proximity regularizer to encourage minimal changes, combining both $L_1$ and $L_2$ distances to promote similarity between the counterfactual and the original input. However, because our framework performs optimization in the latent space and then decodes the perturbed latent vector back to the input domain, the decoder naturally spreads perturbations across the whole time steps. This effect—also observed in latent-space methods such as Glacier—can lead to higher aggregate L1/L2 distances even when the underlying latent perturbations are small. This behavior reflects an inherent trade-off in latent-space approaches: they often produce broader but smoother adjustments rather than very localized changes.  On the other hand, these smoother, distributed perturbations often produce counterfactuals that are more temporally coherent and less abrupt—one of the motivations for adopting a latent-space formulation. As discussed in Section 5.2, this behavior is reflected in the competitive $L_{\infty}$ performance and the favorable plausibility scores, both of which indicate that the generated counterfactuals avoid sharp, unrealistic local changes even when the overall L1/L2 distances are larger. We appreciate the reviewer for highlighting this point; it accurately reflects a fundamental trade-off of latent-space counterfactual generation.
>
> **W3.a(additional per-dataset tables)**: Thank you for the helpful suggestion. We would like to clarify that the submission already includes additional per-dataset tables (**Appendix A.8**), which report all evaluation metrics for every dataset individually. To further strengthen the analysis, we have also added a brief case study on the challenging FordB dataset in Appendix A.8, highlighted in blue, discussing why this dataset is more difficult and how it affects performance. We hope these additions make the empirical results clearer and more comprehensive.
>
> **Q3 and W3.a (clipping)** Thank you for pointing this out. In the original submission, Appendix A.4 (Now A.5) mentioned that the encoder outputs a tanh-bounded latent representation in [−1,1], but did not clearly explain why we also apply hard clipping during counterfactual generation. We have now added a detailed explanation in Appendix A.5 (**highlighted in blue, page 14**).
>
> **W3.a (ablation study)**
> Thank you for this insightful comment. In the revised version, we have added an ablation study that evaluates the contribution of experience replay by comparing the full RL method with a no-replay variant (**see Appendix A.3, and Figure 6, pages 13-14**). The results clarify the role of experience replay: while it is not strictly required for achieving label flips, it plays an important role in generating plausible and data-consistent counterfactuals.
>
> **W3.b**
> Thank you for the helpful suggestion. In the revised version, we added an Efficiency Analysis and Running-Time Comparison in **Appendix A.9 (starting from page 16)**.
>
> **W3.c**: Thank you for the helpful suggestion. In the revised version, we have added a new subsection in the appendix, **A.10 Case Study (page 21)**, where we provide visualizable case studies on two representative datasets. We hope this addition offers a clear and informative qualitative comparison that complements our quantitative results.

---

> > ### Author Response · Authors · 2025-11-23
> > **Continued response to Reviewer 4v39**
> >
> > **W4**: Fixed, thanks for pointing this out!
> >
> > **Q1**: We appreciate the reviewer’s perspective. We did not evaluate a critic-free variant (i.e., using only a single actor network). In our formulation, the critic plays an essential role by providing the actor with a learned assessment of each latent update—capturing how well a candidate counterfactual balances validity, proximity, and plausibility. This value estimate offers stable and informative feedback during training, which is crucial given that the underlying classifier is treated as a black box. A direct actor-only mapping would remove this evaluation signal, forcing the actor to optimize against a highly non-smooth objective without guidance, and would likely lead to unstable training or unrealistic counterfactuals.
> >
> > **Q2**: Thank you for pointing this out. You are correct that in our implementation, the policy directly outputs a perturbed latent counterfactual. The earlier phrasing “action as a perturbation applied to the state’’ was misleading. To avoid confusion, we have revised the text to make the definition consistent with the algorithm: the policy outputs a candidate latent CF. We appreciate the reviewer for catching this inconsistency, and we have updated the manuscript to ensure the terminology is now aligned throughout.

---

### Official Review · Reviewer_nTcX · 2025-10-31

**Soundness:** 2
**Presentation:** 2
**Contribution:** 2
**Rating:** 2
**Confidence:** 4

**Summary:**

This paper works on using RL to generate counterfactual inputs for time series classification tasks. The authors proposed to use DDPG that perturbs inputs in the latent space of a pretrained autoencoder, then decodes them back to the input space. The method is model-agnostic which only needs access to model predictions and operates entirely in latent space. The paper highlights four contributions: (1) an RL approach that scales via batch generation; (2) latent-space perturbations with proximity penalties to promote realism; (3) black-box applicability; and (4) opening RL techniques to counterfactual explanation

**Strengths:**

1. The paper introduces a novel approach that applies reinforcement learning to generate counterfactual explanations for time series data, which is an original and meaningful contribution to the field.
2. The method is thoroughly evaluated on a large number of UCR time series datasets, providing strong empirical support for its effectiveness and generalizability.

**Weaknesses:**

1. The first section of the paper reads as if it may have been generated by an LLM, with generic phrasing and limited depth in motivation and related work discussion.
2. From Figures 2 and 3, the performance improvement of the proposed method over existing approaches appears modest, raising concerns about the overall effectiveness of the method.
3. The paper lacks a runtime analysis comparing the training and inference time of the RL-based approach with prior counterfactual generation methods, which is important for understanding its practical efficiency.
4. The visual presentation could be improved. A clearer overview diagram of the proposed framework or a visual comparison between the RL-based approach and prior methods would greatly help readers understand the system design and key differences.

**Questions:**

1. Could the authors provide a runtime analysis comparing the training and inference times of the proposed RL-based approach with existing counterfactual explanation methods?
2. What is the main motivation for using reinforcement learning in this setting, and why do the authors believe RL is particularly well suited for generating counterfactual time series?
3. Have the authors considered evaluating the method on datasets beyond the UCR collection to further demonstrate its generalizability?

---

> ### Author Response · Authors · 2025-11-23
> **Response to Reviewer nTcX**
>
> **W1, W4, and Q2**: Thank you for these helpful suggestions. We appreciate the reviewer’s concerns regarding the clarity and depth of the introduction, the motivation for using reinforcement learning, and the visual presentation of the proposed framework. To address these points, we have made the following revisions in the updated manuscript: **(1)** We rewrote the first section to provide a deeper and clearer motivation for counterfactual explanations in time series. The revised introduction (**section 1**) now explicitly states that RL allows the agent to learn from experience across many examples, capturing shared temporal patterns instead of re-optimizing for each instance independently. This directly addresses Question 2, explaining that RL is motivated by (a) scalability, (b) generalization, and (c) the ability to learn reusable policies—key advantages not available in instance-based methods. **(2)** Following the reviewer’s suggestion, we added: **Figure 1 (page 2)**: a direct visual comparison between our RL approach and prior instance-based optimization methods, and **Figure 2 (page 6)**: an improved overview diagram of our RL-based counterfactual generation framework. These figures help clarify our framework design, highlight the limitations of instance-based counterfactual methods, and illustrate why learning a transferable policy through RL is advantageous. Together, these revisions improve the motivation, clarity, and visual presentation of the method, and we hope they address the reviewer’s concerns.
>
> **W2**: We agree with the reviewer that in Figure 2 (**now Figure 4**), our RL method does not achieve the smallest L1 or L2 distances. This is expected and consistent with the design of our approach: Our RL method and Glacier perturb the latent representations instead of directly manipulating the raw time series. After decoding, latent-space perturbations naturally produce distributed, global changes across the sequence, yielding higher aggregate L1/L2 distances. This behavior is also described in our results section (**see page 9, section 5.2**). We also show that despite higher aggregate distances, RL performs competitively on $L_{\infty}$, indicating that it avoids extreme spikes or unrealistic local jumps. Thus, RL produces broader but smoother perturbations, which is an intentional design trade-off. And Figure 3 (**now Figure 5**) reveals where RL provides the strongest value: RL achieves the lowest or near-lowest scores under all three OOD detectors (IF, LOF, OC-SVM). Scores remain consistently with lower variance than baselines. Competing methods span a much wider outlier range (up to 1.0), indicating substantially less stable plausibility. This shows that RL’s globally coherent latent-space perturbations produce realistic and in-distribution counterfactuals across 13 datasets—one of the core goals of our proposed method.
> In summary, while RL does not achieve the lowest L1/L2 proximity scores (as expected from latent-space perturbation), it delivers strong and consistent plausibility across all detectors, smooth and non-spiky perturbations (competitive $L_{\infty}$). Thus, the improvements are not modest—rather, they reflect a different performance focus: plausible and realistic time series counterfactuals, supported by strong plausibility and policy generalization.
>
> **W3 and Q1**: Thank you very much for this good suggestion. We agree that understanding the training and inference efficiency of counterfactual methods is important for practical deployment. In response to your suggestion, we have added a new runtime analysis section in the revised manuscript (Appendix Section A.9: Efficiency Analysis and Running-Time Comparison, page 16, starting from line 861), which now provides a detailed comparison across our RL-based method and existing instance-based approaches.
>
> **Q3**: We sincerely thank the reviewer for this thoughtful question. We agree that evaluating counterfactual explanation methods on additional time‐series domains is an important direction for demonstrating broader generalizability. In this work, we focused on UCR because it is the standard benchmark used by all existing counterfactual baselines in the time series domain, which ensures a fair and consistent comparison. That said, we fully agree that extending evaluation beyond UCR-such as to multivariate, irregular, or domain-specific datasets, is valuable. While this is beyond the scope of the current submission, we view this as a natural and impactful avenue for future work. We have updated the revised manuscript accordingly and added a note (**highlighted in red in the Conclusion section, page 10**) to explicitly emphasize that evaluating the method on datasets beyond UCR is an important direction for future work. We greatly appreciate the reviewer’s feedback, which helped us strengthen the discussion and highlight promising directions for improving the generalizability of our method.

---

### Official Review · Reviewer_4wVH · 2025-11-01

**Soundness:** 2
**Presentation:** 2
**Contribution:** 2
**Rating:** 4
**Confidence:** 3

**Summary:**

This paper proposes a reinforcement learning (RL) framework for generating counterfactual explanations (CFEs) in time series classification using batches.  The authors use an actor–critic RL agent to learn a policy in the latent space of a pre-trained autoencoder.
This enables the generation of counterfactuals that are both valid (flip the classifier’s prediction) and  plausible (remain close to the data manifold) without handcrafted constraints. Once trained, the RL agent can generate counterfactuals in batches, making the approach scalable to large datasets. Experiments on 13 UCR benchmark datasets showed reasonable results. The approach is model-agnostic, requiring only access to classifier predictions.

**Strengths:**

- The method is model-agnostic;
- metrics and their relevant are well defined;
- Experiments adopted standard datasets, and reasonable metrics with relevant baselines.

**Weaknesses:**

- Autoencoder performance and its impact on results is not discussed;
- RL to predict Counterfactuals in time-series is not really a novelty, limiting the contribution to the batch predictions. '
(more on the questions)

**Questions:**

- Did you created visualization (or other analysis) on the time-series to illustrate if the timing (of flip) is preserved?
- How much can the autoencoder impact the overall method's performance?
- Are there any datasets where the model performance is worse than expected? (Violin plots shows
comparison with others, wondering if there are any datasets where the method is not performing well)
- How does it compare with methods that don't predict using the batch approach? what is the trade-off between
efficiency (at using batches versus single estimations) and the metrics?

---

> ### Author Response · Authors · 2025-11-23
> **Response to Reviewer 4wVH**
>
> **W1 and Q2**: Thank you for pointing this out. We agree that the quality of the autoencoder is important because our RL agent operates fully in its latent space. To clarify this, we have added a new paragraph in **Appendix A.5 (highlighted in blue, starting from line 771)** explicitly discussing how autoencoder reconstruction quality affects (1) the structure of the latent manifold, (2) the stability of proximity optimization, and (3) the plausibility of decoded counterfactuals. This additional discussion clarifies how the autoencoder influences the overall framework and highlights its role in shaping the latent space used by the RL agent.
>
> **W2:** While reinforcement learning has been used in prior counterfactual explanation work, these approaches have predominantly focused on tabular data. As discussed in our related work section, the landscape of time-series counterfactual methods is largely dominated by instance-specific optimization, where each counterfactual is generated independently for each input without transferring information across samples. Our contribution is not in proposing RL as a new algorithmic idea, but in adapting an RL-driven formulation to the unique structure of time-series data, where counterfactuals must maintain temporal coherence and avoid unrealistic distortions that accumulate across time. By learning a perturbation policy in a structured latent space, our method enables generalizable, experience-driven counterfactual generation, addressing limitations of existing approaches, such as per-instance optimization cost and lack of transferability. We have clarified this motivation in the Introduction and related work sections.
>
> **Q1**: Thank you for raising this point. We would appreciate a brief clarification to ensure we fully understand the reviewer’s concern. In particular, could the reviewer clarify what is meant by “timing (of flip) is preserved”? Once clarified, we are happy to provide the appropriate analysis.
> If the reviewer is referring to whether the generated counterfactuals preserve the original temporal structure or coherence of the input time series, we have added a qualitative case study in **Appendix A.10**, which illustrates this behavior across two representative datasets. We hope this addition is helpful, and we are happy to further expand the analysis based on the reviewer’s clarification.
>
> **Q3**: Thank you for this great question. In addition to the violin plots, we reported the detailed per-dataset performance for all evaluation metrics in Appendix A.7, which makes it easy to examine how the method behaves on each dataset individually. From these results (Tables 2–8 in Appendix A.8), we did observe that some datasets are more challenging. In particular, FordB shows lower validity compared to others, which we found to be associated with both lower classifier accuracy and a higher autoencoder reconstruction error. To make this clearer, we have added a short case study in Appendix A.8 (**highlighted in red in the revision, starting from line 849**), analyzing FordB and explaining why counterfactual generation is more difficult on this dataset. We hope this additional analysis provides a clear understanding of dataset-level variability in performance.
>
> **Q4:** Thank you for the helpful suggestion. We have added a clear comparison in the revised version, including an Efficiency Analysis and Running-Time Comparison in **Appendix A.9 (starting from lines 861)**. We hope this addition provides a more transparent view of the differences between batch-based generation and single-instance counterfactual methods.

---

### Official Review · Reviewer_fYoZ · 2025-11-02

**Soundness:** 3
**Presentation:** 2
**Contribution:** 3
**Rating:** 6
**Confidence:** 5

**Summary:**

This paper proposes a counterfactual explanation method for time series classification based on a reinforcement learning (RL) framework. Inspired by prior work on tabular data, the authors extend the approach to time series and evaluate it on multiple real-world datasets.

**Strengths:**

Strength:
1 The topic is interesting, and to my knowledge, this is the first work that applies RL to generate counterfactual explanations (CFs) for time series classification.
2 The experiments are abundant.
3 The algorithm1 is clear and well introduced.

**Weaknesses:**

Weakness:
1 Although the paper introduces RL for generating CFs in time series classification, it does not clearly distinguish this task from CF generation in tabular classification. This distinction should be clarified.
2 he generated CFs satisfy the proximity principle through a regularization term, but the paper does not address whether the produced CFs are realistic or plausible in real-world time series scenarios.
3 The writing assumes familiarity with the definition of counterfactuals. For readers who are new to CFs, the lack of a clear formal definition for time series may reduce readability.
4 The related work section does not cite prior RL-based counterfactual generation methods in tabular classification.
5 The method does not discuss or address the challenge of high-dimensional action spaces in time series classification.

**Questions:**

Suggestions and Questions:
1 I suggest adding a formal problem definition section before Section 4, including:
    1.1 A clear mathematical formulation of counterfactuals in time series classification.
     1.2 A comparison with the tabular setting, highlighting what makes the time series case different.
2 The related work section should be expanded. There are several well-known papers that apply RL to counterfactual generation in tabular data; I list some of them below for reference:
   2.1 Chen, Ziheng, Fabrizio Silvestri, Jia Wang, He Zhu, Hongshik Ahn, and Gabriele Tolomei. "Relax: Reinforcement learning agent explainer for arbitrary predictive models." In Proceedings of the 31st ACM international conference on information & knowledge management, pp. 252-261. 2022.
   2.2 Nguyen, Tri Minh, Thomas P. Quinn, Thin Nguyen, and Truyen Tran. "Counterfactual explanation with multi-agent reinforcement learning for drug target prediction." arXiv preprint arXiv:2103.12983 (2021).

3 I have a question for formulat 4. As the action is taken step by step and it is easy to control the sparsity of the modified features. Why do you optime the L1 norm instead of the L0 norm directly?

---

> ### Author Response · Authors · 2025-11-23
> **Response to Reviewer fYoZ**
>
> **W1 and Q1.2**: Thank you for the insightful comment. We agree that clearly distinguishing counterfactual generation for time series from the tabular setting is important, and we have revised the Introduction to make this distinction explicit. This expanded explanation (included in the revised manuscript, **highlighted in red in Section 1 beginning at line 48**) directly addresses the reviewer’s concern by clarifying why counterfactual generation for time series is qualitatively different from the tabular setting and why an RL-based approach is particularly well-motivated for this problem.
>
> **W2**. Thank you for highlighting this point—we appreciate the opportunity to clarify it. We agree that plausibility is a crucial part of counterfactual quality, especially for time series. In our original submission, we already evaluated plausibility using several standard OOD-based metrics (LOF, Isolation Forest, and One-Class SVM), which assess whether the generated counterfactuals stay close to the underlying data manifold. These evaluations and results are reported in **Section 5.2 and in Tables 6–8 of Appendix A.8**, and our method shows consistently strong plausibility across 13 datasets. That said, we understand that our original explanation may not have made the role of these metrics sufficiently clear. To improve readability, we have refined the plausibility evaluation metric description (**at the end of section 5.1 on page 9, where the refined text is highlighted in blue**) to explicitly note that each OOD detector is trained on the original training data to model the normal time series distribution, and that the outlier scores of the generated counterfactuals serve as quantitative measures of realism in real-world scenarios.
>
> **W3 and Q1.1**. Thank you for pointing this out. We agree that including a formal problem definition is important for readers who may be new to counterfactual explanations. To improve clarity, we have added an explicit mathematical formulation of counterfactual generation for time series classification in **Section 3.1**.
>
> **W4 and Q2**: Thank you for this valuable suggestion. We agree that existing RL-based counterfactual methods in tabular classification are highly relevant and should be acknowledged. In the revised manuscript, we have expanded the Related Work (**section 2 highlighted in red, starting at line 158**) to include these approaches, including the RL formulations by Samoilescu et al. (2021), the RELAX method by Chen et al. (CIKM 2022), and the multi-agent RL framework by Nguyen et al. (2021). We also added a short discussion explaining that these works highlight the promise of experience-driven policy learning for counterfactual generation in tabular domains and motivated our exploration of RL for time series. The revised text now clearly situates our contribution within this line of research while emphasizing that RL-based counterfactual generation has been largely unexplored for time series. We appreciate the reviewer for pointing us to these important references, which helped strengthen the completeness of our Related Work section.
>
> **W5**: Thank you for raising this insightful point. We agree that high-dimensional action spaces present a significant challenge in time series counterfactual generation. Our method addresses this issue by performing all RL operations in a low-dimensional latent space rather than the raw time-series domain. Specifically, as described in Section 4, we first train an autoencoder to obtain a compact latent representation of the input sequence. The RL agent operates exclusively in this latent space, which greatly reduces the dimensionality of the action space. To make this design choice clearer, we have refined the corresponding paragraph in **Section 4 (highlighted in blue), starting from line 227** to explicitly state that operating in the latent space mitigates the curse of dimensionality that would arise from perturbing every time step of the raw sequence. We hope this clarification helps highlight how the method effectively handles the high-dimensional nature of time series inputs.
>
> **Q3**: Thank you for the question. While the $L_0$ norm directly measures sparsity, it is non-differentiable and cannot be optimized using gradient-based updates. Since our actor–critic framework relies on backpropagation for policy learning, we use the $L_1$ norm as its standard differentiable relaxation, which is common in counterfactual explanation methods (e.g., Wachter et al., 2017; Mothilal et al., 2020). This allows us to encourage sparse perturbations while maintaining stable training.

---

### Meta-Review · Area_Chair_jCpy · 2026-01-06

**Summary:**

The submission presents an RL-based framework for generating counterfactual explanations for time-series classifiers. Reviewers raised concerns primarily about

(i) Reviewer `fYoZ`, Reviewer `4wVH`: insufficient positioning with respect to prior RL-based counterfactual methods in tabular settings and an unclear articulation of what is fundamentally different in the time-series case;

(ii) Reviewer `4v39`: conceptual clarity of the RL formulation, in particular the one-step horizon and whether the problem is better viewed as a contextual bandit rather than a sequential MDP, raising questions about the necessity of a full actor–critic framework;

(iii) Reviewer `4v39`: the trade-off between plausibility and minimality, as the method often produces counterfactuals with larger L1/L2 distances compared to optimization-based baselines;

(iv) Reviewer `nTcX`, Reviewer `4v39`: missing empirical analyses in the original submission, including runtime comparisons, ablations, and qualitative examples.

While the rebuttal and revision improve clarity and address several experimental gaps, skepticism remains regarding the strength of the novelty claim and the need for the proposed level of RL complexity.

**Reviewer Concerns:**

## Concerns addressed by the rebuttal

1. **Positioning and missing related work**:
The authors expanded the related work to include prior RL-based counterfactual methods and clarified the intended novelty relative to tabular settings.

2. **Formal definition and readability**:
A formal mathematical formulation for time-series counterfactual generation was added, improving accessibility for readers unfamiliar with counterfactual explanations.

3. **Empirical gaps**:
The revision added an efficiency/running-time comparison, per-dataset result tables and a dataset-specific case study, a qualitative case-study section, and an ablation on experience replay.

4. **RL framing inconsistency**:
The authors acknowledged and corrected the earlier sequential wording, explicitly reframing the method as a contextual bandit / one-step MDP and clarifying the critic’s role as an immediate reward estimator.


## Concerns still outstanding

1. **Strength of novelty and contribution**:
Despite improved positioning, the novelty-related issues are not fully resolved in the response, particularly given that the formulation is ultimately clarified as a single-step decision problem rather than a sequential process.

2. **Necessity of actor–critic complexity in a one-step setting**:
While the paper now correctly describes the setting as contextual bandit, the justification for using a DDPG-style actor–critic (vs. simpler actor-only/bandit or direct optimization) remains largely qualitative. The authors did not evaluate a critic-free variant, leaving the complexity-benefit trade-off not fully substantiated.

3. **Minimality / actionability trade-off**:
The authors explain that latent-space perturbations can produce smoother yet more globally distributed changes after decoding, which may increase L1/L2 distances. However, the concern that larger distances can conflict with minimal change goals is not fully resolved beyond a trade-off argument.

4. **Generality beyond UCR**:
The rebuttal indicates that evaluation beyond UCR is future work; thus, external generalizability remains unverified.

**Reviewer Scores:**

**Reviewer `fYoZ` (Score: 6, marginally above acceptance)**

The rebuttal directly addressed most of this reviewer’s concerns. This reviewer would likely maintain the score around 6.

**Reviewer `4wVH` (Score: 4, marginally below acceptance)**

The authors added discussion during the rebuttal period, which addresses several of the reviewer’s technical questions. However, the response does not fully resolve the concern regarding the level of novelty relative to existing RL-based counterfactual approaches. As a result, a slight positive adjustment is possible, but the assessment would likely remain in the borderline range.

**Reviewer `nTcX` (Score: 2, reject)**

The revision addresses some of the reviewer’s points. However, the core concern in the original review relates to limited performance gains and a perceived weak overall contribution, which are not substantially altered by the rebuttal. Therefore, only a modest change, if any, would be plausible, and the evaluation would likely remain negative.

**Reviewer `4v39` (Score: 4, marginally below acceptance)**

The rebuttal corrected RL framing (contextual bandit) and addressed several major concerns. Nonetheless, the response does not fully resolve questions regarding the necessity of the actor–critic complexity in a one-step setting or the trade-off between proximity and minimality. Given these remaining issues, it is plausible that the score would remain around 4.

---

### Decision · Program_Chairs · 2026-01-26

Reject